# MacroD1 sustains mitochondrial integrity and oxidative metabolism

Ann-Katrin Hopp [1,2], Lorenza P. Ferretti[1], Lisa Schlicker [3], Amalia Ruiz-Serrano[4], Udo Hetzel[5], Francesco Prisco [6], Anja Kipar [6], Lukas Muskalla[1,2], Elena Ferrari[1], Carsten C. Scholz [4,7], Karsten Hiller [3], Francisco Verdeguer[1], Deena M. Leslie Pedrioli [1] & Michael O. Hottiger [1] ✉

The mono-ADP-ribosylhydrolase MacroD1 has been recently reported to localize to mitochondria exclusively. However, the extent and means by which MacroD1 regulates metabolic homeostasis remains unclear. Here we show that the absence of MacroD1 in mice decreased mitochondrial load and negatively impacted muscle function, reducing maximal exercise capacity. Knockdown of MacroD1 in C2C12 myoblast cells amplified the production of reactive oxygen species which ultimately resulted in increased mitochondrial fission. Proteomic and metabolomic profiling showed that loss of MacroD1 re-routed metabolite flux from glucose to the pentose-phosphate cycle instead of the tricarboxylic acid cycle to support the production of antioxidants, including glutathione and NADPH. This resulted in increased glucose uptake and dependency both in vitro and in vivo. Hence, our research establishes MacroD1 as a regulator of metabolic homeostasis, which ensures the coordination of cellular carbohydrate flux and optimal mitochondrial function.

Mitochondria are the cells' main energy hub, converting consumed and metabolized nutrients, like pyruvate, from sugar via TCA cycle activity and oxidative phosphorylation (OXPHOS) into ATP or GTP[1]. In addition, pyruvate can also be metabolized by lactate dehydrogenase (LDH), which results in the generation of cytoplasmic lactate and the regeneration of cytoplasmic nicotinamide adenine dinucleotide (NAD+). The fate of pyruvate, thus, depends on oxygen and nutrient availability, mitochondrial respiratory capacity, and the energetic demand of the cell or organ[2]. Mitochondria and the cytoplasm have distinct requirements for NAD+, and proper compartmentalization of redox equivalents is crucial for maintaining cellular homeostasis and survival in response to environmental stressors[3–5]. While OXPHOS yields higher energy production from each glucose unit compared to anaerobic glycolysis, the latter is kinetically faster. The choice of OXPHOS versus lactate-dependent energy production involves multiple regulatory mechanisms that are still not fully understood. These metabolic fluxes are, for example, not only dependent on oxygen availability as observed by the Warburg effect[2] but are also linked to mitochondrial morphology and dynamics, such as their fusion or fission events[6]. Mitochondrial fusion is positively associated with increased ATP production, while inhibition of fusion is related to impaired OXPHOS, mitochondrial DNA depletion, and/or ROS production[6,7].

MacroD1 (also referred to as leukemia-related protein 16; LRP16) is 1 of at least 12 different macrodomain-containing proteins identified in humans to date[8]. Catalytically active macrodomains are involved in the turnover of small metabolites, such as ADP-ribose (ADPr) or ADPr adducts, like O-acetyl-ADPr (OAADPr) or ADP-ribose-1"-phosphate, while macrodomains lacking catalytic activity often serve to bind to (poly-) ribonucleotides like DNA, RNA, poly- or mono-ADP-ribose (PAR

[1]Department of Molecular Mechanisms of Disease (DMMD), University of Zurich, Zurich, Switzerland. [2]Life Science Zurich Graduate School; Molecular Life Science Ph.D. Program, Zurich, Switzerland. [3]Institute for Biochemistry, Biotechnology and Bioinformatics, Braunschweig Integrated Centre of Systems Biology (BRICS), Braunschweig, Germany. [4]Institute of Physiology; University of Zurich, Zurich, Switzerland. [5]Electron Microscopy Unit, Institute of Veterinary Pathology, University of Zurich, Zurich, Switzerland. [6]Laboratory for Animal Model Pathology (LAMP), Institute of Veterinary Pathology, University of Zurich, Zurich, Switzerland. [7]Institute of Physiology, University Medicine Greifswald, Greifswald, Germany. ✉e-mail: michael.hottiger@dmmd.uzh.ch

and MAR, respectively)[9–16]. Individual macrodomains differ widely regarding substrate binding affinity, specificity, and enzymatic activity[14]. These differences explain why macrodomain-containing proteins can exert very different functions and are involved in various biological processes, including chromatin remodeling, transcriptional regulation, or host-pathogen interactions[17–20]. Interestingly, two recent studies demonstrated that both endogenous and overexpressed MacroD1 localize predominantly to mitochondria[14,21,22] and that MacroD1 expression positively correlates with the cellular mitochondrial load[21,22]. In line with that, in WT mice, MacroD1 expression levels in skeletal and heart muscle have been reported to be very high[21]. Skeletal muscle tissue is rich in mitochondria and is not only a vital nutrient storage organ but also one of the most dominant glucose consumers. Depending on muscle type and function, glucose is either used to fuel anaerobic or aerobic metabolism or stored as glycogen. Because of its dual role, it is unsurprising that, together with the liver, skeletal muscle is an integral part of whole-body physiology and blood glucose homeostasis[23]. Despite its high abundance, the role of MacroD1 in skeletal muscle, as well as its impact on mitochondrial metabolism, has thus far remained elusive.

Here, we provide evidence that MacroD1 is an important regulator of the mitochondrial structure and function. In vitro and in vivo functional assays, combined with proteomic and metabolomic studies, revealed that loss of MacroD1 resulted in irreversible mitochondrial damage. This damage led to a concomitant metabolic shift from OXPHOS to the pentose-phosphate pathway (PPP), leading to enhanced glucose uptake and increased insulin sensitivity in vivo. Taken together, we identify MacroD1 as an important regulator of mitochondrial function and metabolism.

## Results

### MacroD1 is essential for oxidative metabolism in muscle

To gain new insights into the physiological role of MacroD1, we first characterized the impact of a whole-body MacroD1 knockout in C57BL/6N mice (MacroD1$^{-/-}$). Given that MacroD1 has been proposed to be a negative regulator of insulin signaling[24], the body weight of wildtype and MacroD1$^{-/-}$ mice on standard chow diet was monitored for 11 weeks. No difference in weight gain was observed in MacroD1$^{-/-}$ compared to wildtype animals, neither in females nor in males (Fig. 1A and Supplementary Fig. 1A). To further confirm that loss of MacroD1 did not alter feeding behavior, different parameters such as body weight, food and water intake or excretion were monitored for male and female MacroD1$^{-/-}$ and wildtype animals for three days. None of the tested parameters were altered in MacroD1$^{-/-}$ mice (Supplementary Fig. 1B–E), indicating that sustained loss of MacroD1 did not alter body weight or feeding behaviors under the conditions tested here.

To determine if loss of MacroD1 affects body mass distribution, the body composition of male and female MacroD1$^{-/-}$ mice was analyzed by EchoMRI. Interestingly, we observed a significant increase in the fat mass of male MacroD1$^{-/-}$ animals compared to male wildtype mice and a similar trend in female animals (Fig. 1B) as well as a slight but statistically not significant decrease in lean mass (Supplementary Fig. 1F), while free and total fluidics were comparable between wildtype and MacroD1$^{-/-}$ mice (Supplementary Fig. 1G, H). To detect other potential phenotypic changes associated with the loss of MacroD1, a histological examination of a complete set of organs and tissues from three MacroD1$^{-/-}$ and two age-matched wildtype mice (6 weeks old) was performed. However, no macroscopic or histological abnormalities were found in MacroD1$^{-/-}$ mice.

As MacroD1 is highly expressed in skeletal muscle[21,22], MacroD1$^{-/-}$ animals were subjected to a muscle performance test consisting of involuntary running until exhaustion. While the total running time recorded for MacroD1$^{-/-}$ mice was only marginally reduced (Fig. 1C), their respiratory exchange ratio (RER) was consistently higher compared to wildtype animals (Fig. 1D). In fact, MacroD1$^{-/-}$ mice reached RER values above 0.9 significantly earlier than wildtype animals (Fig. 1D, dashed line). These findings suggest that the maximal exercise capacity of MacroD1$^{-/-}$ mice is reduced and that they are more dependent on carbohydrates as a fuel/energy source. Interestingly, the RER of MacroD1$^{-/-}$ animals was already significantly higher under basal conditions as compared to wildtype animals, suggesting a decrease in fat and/or an increase in glucose utilization in the muscle of MacroD1$^{-/-}$ animals. In agreement with this conclusion, oxygen consumption in MacroD1$^{-/-}$ mice was significantly lower than in wildtype animals (Fig. 1E). Taken together, these findings suggest that loss of MacroD1 reduces oxidative metabolism in mouse muscle.

To strengthen this point, the expression of genes involved in mitochondrial OXPHOS was analyzed by qPCR in isolated hindlimb skeletal muscles from MacroD1$^{-/-}$ and wildtype mice. Indeed, the expression of genes involved in OXPHOS (e.g., *Ndufa9* and *Err1*) was found to be significantly lower in MacroD1$^{-/-}$ muscle (Fig. 1F). To determine whether this observation was muscle-specific or a general characteristic of mitochondria-rich tissues, we also analyzed brown adipose tissue (BAT), another energy end consumer. Similar to the expression profiles observed in muscle, the expression levels of several genes involved in mitochondrial OXPHOS were also found to be significantly lower in BAT isolated from MacroD1$^{-/-}$ mice (Supplementary Fig. 1I). These findings suggest that MacroD1 is not only important for oxidative metabolism in skeletal muscle but also in other metabolically active tissues.

Considering the effect of MacroD1 deficiency on exercise capacity, we assessed if this was associated with overt subcellular changes or a decrease in myofiber size and, therefore, undertook an ultrastructural examination of the *Musculus (M.) quadriceps femoris*, diaphragm and myocardium in a group of male MacroD1$^{-/-}$ and wildtype mice (Fig. 2A). Although there was no evidence of myofiber damage, the average Z line lengths were significantly shorter in all samples of MacroD1$^{-/-}$ mice compared to wildtype animals, indicating that the diameter of sarcomeres is smaller in the male MacroD1$^{-/-}$ mice (Fig. 2A). Subsequently, an in-depth in situ examination was undertaken on the *M. quadriceps femoris*, *M. triceps brachii*, *M. gastrocnemius*, and *M. soleus* of 10 MacroD1$^{-/-}$ and wildtype mice (5 males and 5 females each). This histological examination did not reveal any morphological differences between MacroD1$^{-/-}$ and wildtype mice (Supplementary Fig. 1J). Nevertheless, the comparative assessment of muscle fiber diameters showed that in MacroD1$^{-/-}$ mice both the *M. quadriceps femoris* and *M. triceps brachii* fibers were significantly smaller in diameter than in the controls ($p < 0.0001$ for both muscles), while there were no significant differences in fiber diameter in the *M. gastrocnemius* and *M. soleus* ($p = 0.0566$ and $p = 0.2345$ respectively) (Fig. 2B). While the reduction in muscle fiber diameter in *M. quadriceps femoris* and *M. triceps brachii* was observed independently of the animals' sex, the difference was more pronounced and highly significant in female mice (Supplementary Fig. 2A). Immunohistochemistry for type I myofiber-specific myosin performed on the *M. soleus*, which is known to generally exhibit an approximately even distribution of type I and II myofibers in mice[25], showed a significantly higher proportion of type II fibers (fast-twitch, glycolytic fibers) in MacroD1$^{-/-}$ compared to wildtype mice ($p < 0.0001$, Fig. 2C).

### MacroD1 maintains mitochondrial integrity and structure in muscles

To further investigate the effect of MacroD1 loss on muscle mitochondria, the *M. triceps brachii* was stained for enzymatic activity of succinic acid dehydrogenase (SDH) as a mitochondrial marker. The intensity of the SDH activity staining was lower, however not significant in muscles from MacroD1$^{-/-}$ mice, suggesting that the overall oxidative capacity of the muscles might be lower in these mice compared to the wildtype mice ($p = 0.6108$; Fig. 2D and Supplementary Fig. 2B). Furthermore, comparison of the overall mitochondrial load

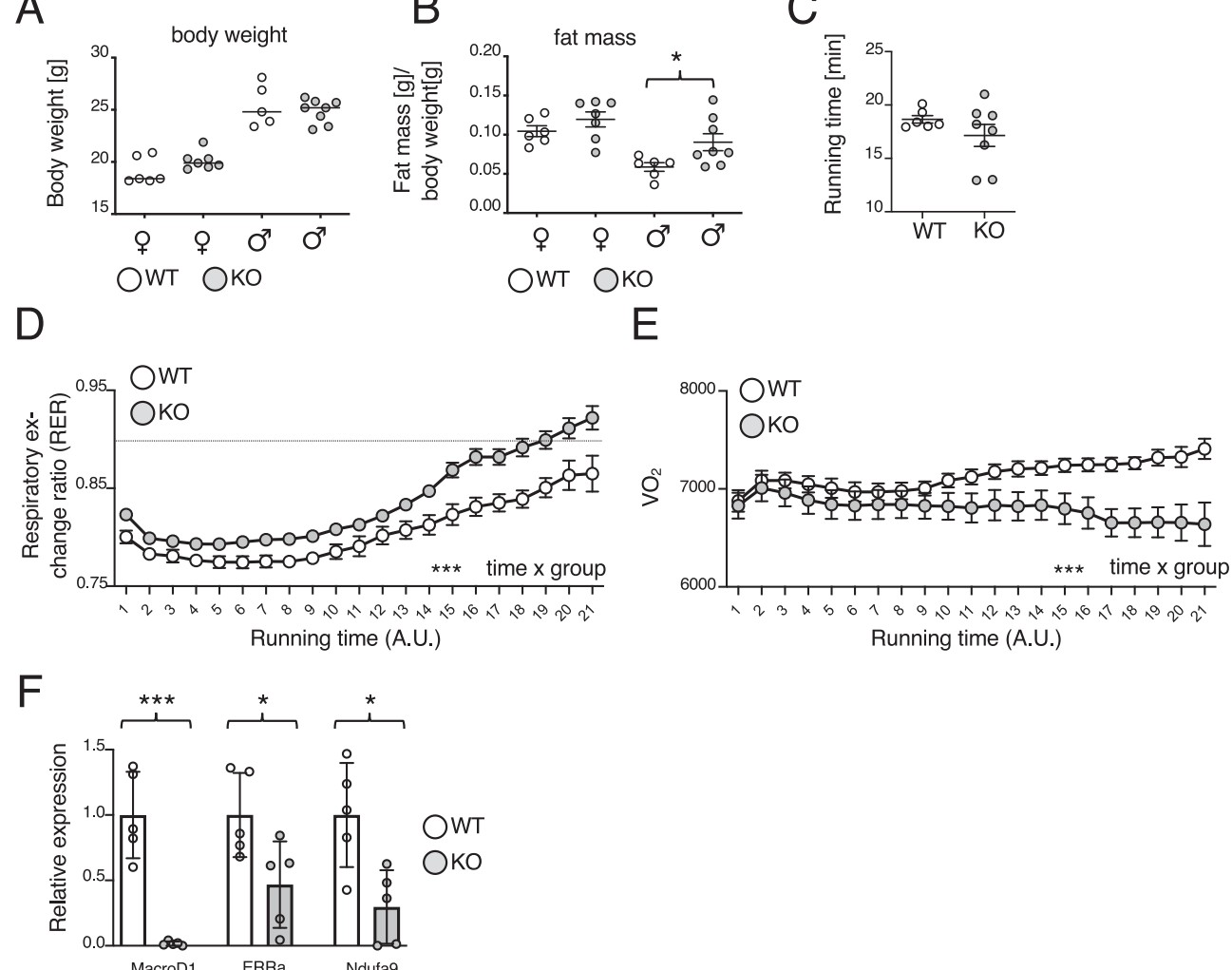

**Fig. 1 | Lack of MacroD1 decreases oxidative metabolism in mouse muscle.**
**A** Weight gain of MacroD1$^{-/-}$ (KO) and wildtype (WT) male (♂) and female (♀) mice on a standard chow diet was monitored weekly over the course of 2 months. Respective weight measures at week 2 are shown as an example. **B** Fat mass of female (♀) and male (♂) MacroD1$^{-/-}$ (KO) and wildtype (WT) animals. Data from $n = 6–8$ individual mice is shown with the line indicating the median. Statistical analyses were performed using a two-tailed student t-test (*, $p < 0.05$; **, $p < 0.005$; ***, $p < 0.0005$). **C–E** Age-matched male MacroD1$^{-/-}$ (KO) and wildtype (WT) mice were submitted to involuntary running until exhaustion, and total running time (**C**), respiratory exchange ratio (RER) (**D**), and oxygen consumption (**E**) were measured. Data is shown as mean ± SD from $n = 6–8$ individual mice. For statistical analysis, a two-way Anova with Geisser-Greenhouse correction as performed (*, $p < 0.05$; **, $p < 0.005$; ***, $p < 0.0005$) on data from $n = 6–8$ individual mice. **F** Gene expression analysis on muscle lysates from MacroD1$^{-/-}$ and wildtype mice. Data is shown as mean ± SD from $n = 5$ individual mice. Statistical analyses were performed using a two-tailed student t-test (*, $p < 0.05$; **, $p < 0.005$; ***, $p < 0.0005$). Source data are provided as a Source Data file.

between MacroD1$^{-/-}$ and wildtype animals revealed a significant decrease in the number of mitochondria in the myocardium of MacroD1$^{-/-}$ mice (Fig. 2E and Supplementary Fig. 2C). The complementary ultrastructural examination revealed structural deformations in the mitochondrial cristae in the muscles of MacroD1$^{-/-}$ animals (Fig. 2F; exemplary images of myocardium and *M. triceps brachii*, showing mitochondria with altered cristae). Interestingly though, the observed mitochondrial changes were not associated with detectable changes in the amount of mitochondrial DNA (Supplementary Fig. 2D). Together, these data suggest that MacroD1 is important for structural integrity, but not DNA copy number of muscle mitochondria in vivo.

To investigate the molecular mechanism responsible for the observed structural changes in mouse muscle mitochondria, we decided to proceed with the C2C12 cell line as an in vitro muscle cell culture system[26]. To confirm that this system mimics the genetic conditions present in MacroD1$^{-/-}$ muscle, *Macrod1* was knocked down via siRNA (Supplementary Fig. 3A), and mitochondrial integrity and

morphology were analyzed via transmission electron microscopy (TEM). Similar to the observations above describing altered cristae for muscle mitochondria of MacroD1$^{-/-}$ animals (Fig. 2F), knockdown of *Macrod1* in C2C12 cells resulted in distinct structural deformations of the mitochondrial cristae (Fig. 2G). In addition, the knockdown appeared to be associated with increased mitochondrial fission. To confirm this observation, mitochondrial morphology was analyzed via immunofluorescence (IF) staining of the mitochondria localized protein COXIV. While mitochondria from C2C12 cells transfected with a scrambled siRNA were filamentous, *Macrod1* knockdown resulted in extensive mitochondrial fragmentation (i.e., fission) (Fig. 3A and Supplementary Fig. 3B). To obtain a more quantitative and unbiased assessment of the changes in mitochondrial morphology, single cells from multiple micrographs per condition were analyzed in FIJI using the "Mitochondria Analyzer" plugin. Images were first pre-processed to remove background noise and sharpen mitochondria-specific signals (comparison between original and processed images shown in Supplementary Fig. 3C) and various features, including mitochondrial

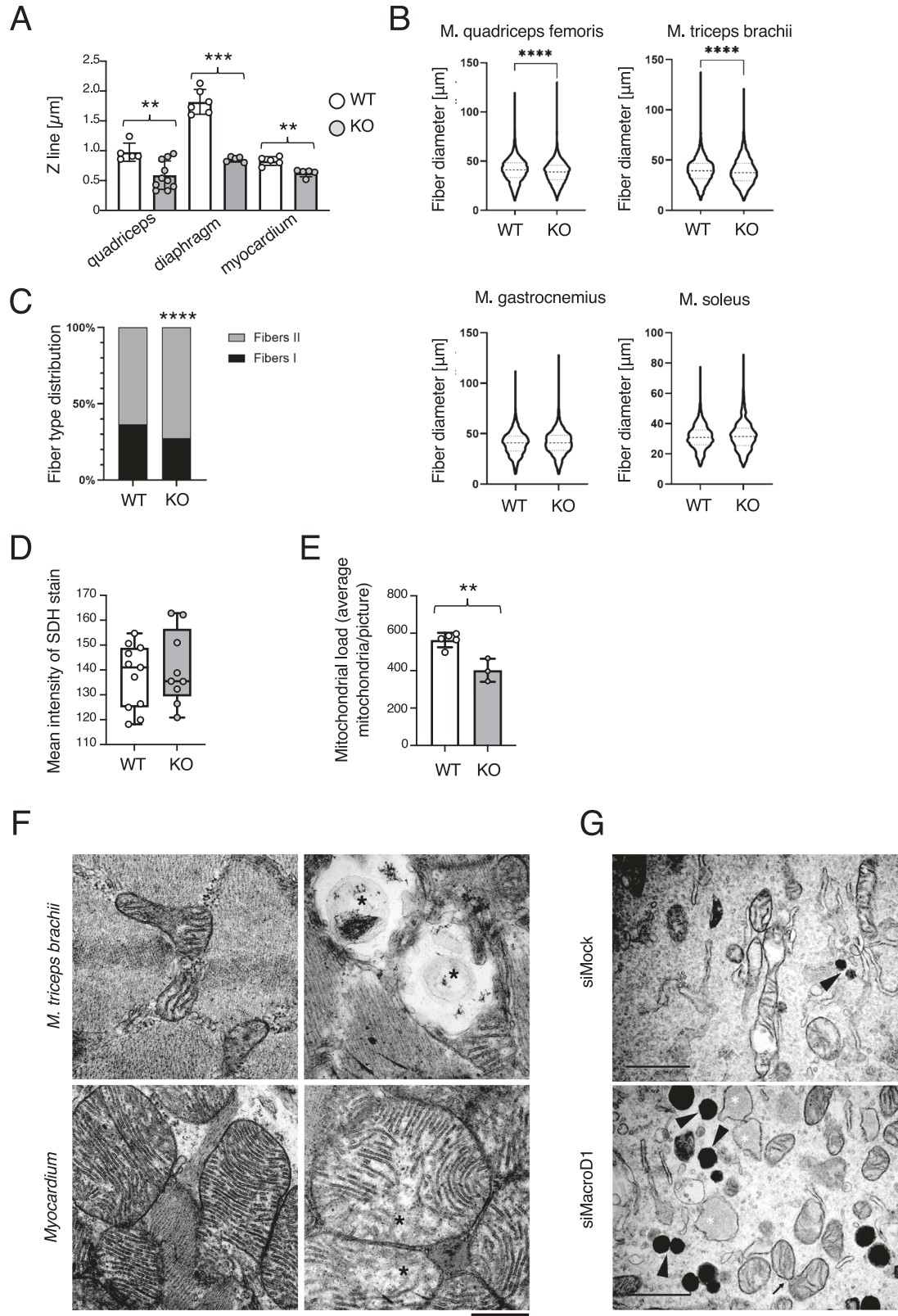

length and branching, were subsequently automatically analyzed. Consistent with the manual assessment, these analyses revealed significant changes in mitochondrial morphology (Fig. 3B, left panel), mean length of branches per mitochondrion (Fig. 3B, mid panel), and the total number of branch end points per mitochondrion per cell (Fig. 3B, right panel). All three parameters were significantly reduced following *Macrod1* knockdown, supporting the idea that loss of

MacroD1 results in shorter and less branched mitochondria, ultimately leading to severe mitochondrial fragmentation. This fission phenotype following *Macrod1* knockdown was observed not only in C2C12, but also in several other human and murine cell lines tested (e.g., HL-1, U2OS and 3T3; Supplementary Fig. 3D–F), suggesting that the observed effect of MacroD1 on mitochondrial structure is general, rather than cell type specific. Taken together, the loss of MacroD1

**Fig. 2 | MacroD1 is relevant for myofiber size and mitochondrial quality.**
**A** Measurement of Z line lengths in ultrastructural images of *Musculus (M.) quadriceps femoris*, diaphragm and myocardium in male MacroD1$^{-/-}$ (KO) and wildtype (WT) mice (each *n* = 4). Data is shown as mean ± SD. Statistical analyses were performed using a two-tailed student t-test (*, *p* < 0.05; **, *p* < 0.005; ***, *p* < 0.0005).
**B** Determination of myofiber diameter (transverse sections) in HE stained cryosections of *M. quadriceps femoris, M. triceps brachii, M. gastrocnemius*, and *M. soleus* of MacroD1$^{-/-}$ (KO) and wildtype (WT) mice (each 5 males and 5 females). Median and quartiles are depicted as dashed lines. For statistical analysis, a Mann–Whitney U test was used (*, *p* < 0.05; **, *p* < 0.005; ***, *p* < 0.0005; ****, *p* < 0.0001).
**C** Immunohistochemistry for type I myofiber-specific myosin, performed on cryosections of the *M. soleus*. A Chi-square test was used for statistical analysis (*, *p* < 0.05; **, *p* < 0.005; ***, *p* < 0.0005; ****, *p* < 0.0001). **D** Quantification of the

staining (mean intensity) for the *M. triceps brachii* for SDH activity (mitochondrial marker). The median is depicted as straight line, whiskers indicate minimal to maximal value. Statistical analyses were performed using a two-tailed student t-test (*, *p* < 0.05; **, *p* < 0.005; ***, *p* < 0.0005). **E** Mitochondrial counts were undertaken in ultrathin sections of the myocardium from MacroD1$^{-/-}$ (KO) and WT mice. **F** TEM images of skeletal muscle (*M. triceps brachii*) and myocardium in MacroD1$^{-/-}$ (KO) and wildtype (WT) mice. In MacroD1$^{-/-}$ mice, mitochondria are swollen and exhibit disarrangement and partial loss of cristae (cristolysis; asterisks). Bar = 500 nm.
**G** TEM images of C2C12 cells. After the knockdown of *Macrod1* (siMacroD1), mitochondria exhibit disarrangement and partial loss of cristae (cristolysis; asterisks). There is evidence of mitochondrial fission (arrow). The arrowheads highlight electron dense cytoplasmic granules present in both siMock and siMacroD1 cells. Bars = 1 μm. Source data are provided as a Source Data file.

reduced mitochondrial cristae and network formation both in vivo and in vitro.

## Extended knockdown of *Macrod1* induces apoptosis in different cell types

In addition to the mitochondrial fission phenotype, we also observed a significant increase in the number of condensed round nuclei, which were smaller in size than an average G1 nucleus, and a reduction in the overall number of cells, starting 2 days after *MacroD1* knockdown and becoming strongly apparent 3 days after knockdown (Supplementary Fig. 3G, H). Since mitochondrial fission is known to affect cell viability[27], we measured cell viability upon *Macrod1* knockdown. Co-staining of AnnexinV with propidium iodide (PI) was performed three days after *MacroD1* knockdown and analyzed via flow cytometry to assess the extent of apoptosis. Indeed, *Macrod1* knockdown resulted in a significant increase in AnnexinV-positive apoptotic cells, while no significant changes in the number of PI-positive, necrotic cells were observed (Fig. 3C). In line with the increase in AnnexinV-positive cells, Western blot analysis further revealed that knockdown of *Macrod1* also increased PARP1 cleavage and caspase 3 protein levels (Supplementary Fig. 3I, J), thus confirming that extended knockdown of *Macrod1* induced apoptotic cell death. A similar increase in AnnexinV-positive cells was also observed in HL-1 cells (Supplementary Fig. 3K). Since most cells died after 3 days (72 h), all further readouts were evaluated between 48 h to 60 h of knockdown.

The critical role of MacroD1 for cell viability seems inconsistent with the fact that MacroD1$^{-/-}$ mice are viable. This discrepancy could be explained by the embryonic stem cells adapting to the loss of MacroD1 during the selection process following the CRISPR/Cas9-mediated deletion. To investigate this possibility, shRNA constructs targeting *Macrod1* were stably transfected into C2C12 cells (Supplementary Fig. 4A). Indeed, while most cells died during the selection process, a small fraction of the cells survived. Together, these findings confirmed that loss of MacroD1 impairs cell viability, but also demonstrated that cells can apparently functionally compensate for the loss of MacroD1 under the conditions tested. However, similar to what we observed after siRNA-mediated transient knockdown of *Macrod1*, stable knockdown of *Macrod1* also resulted in increased mitochondrial fragmentation (Supplementary Fig. 4B) and a significant reduction in mitochondrial load (Supplementary Fig. 4C). To further assess mitochondrial quality following stable knockdown of *Macrod1*, shMacroD1 and control C2C12 cells were either left untreated or treated with FCCP (as control) and then stained with the depolarization-sensitive mitochondrial dye TMRE prior to flow cytometry analysis. Remarkably, shMacroD1 C2C12 cells showed a significant decrease in TMRE signal and, thus, a reduction in mitochondrial polarization and the ability to drive protons in the intermembrane space (Supplementary Fig. 4D), further strengthening the interpretation that MacroD1 plays an important role in regulating mitochondrial morphology and allowing cells to adapt to the loss of MacroD1.

## MacroD1 gene knockdown increases mitochondrial ROS production

To gain further mechanistic insights into what might lead to the above-described mitochondrial fission and ultimate cell death phenotypes, U2OS cells transfected with siRNA targeting *MACROD1* or a scrambled siRNA (*n* = 4/group) were lysed 60 h post-transfection, and the whole cell proteome was analyzed using label-free quantification (LFQ) based mass spectrometry (MS) methodologies[28]. The 60 h time point was chosen, as it represented a compromise between very good knockdown efficiencies without giving the cells much time to adapt. In total, ~3700 quantifiable proteins were identified, and clustering analyses based on Spearman correlation revealed that siMock and siMacroD1 samples clustered together (Supplementary Fig. 5A), indicating that there was very little variation between the biological replicate samples, as well as data robustness and reproducibility. Comparing siMacroD1-treated cells to siMock-treated cells allowed us to identify 128 upregulated proteins and 239 downregulated proteins whose abundances significantly (p-value ≤ 0.05) changed ≥1.5 fold following *MacroD1* knockdown (Supplementary Fig. 5B, Supplementary Data 1). Gene ontology (GO) analysis of all increased proteins (Fig. 3D, Supplementary Fig. 5C) revealed a significant enrichment of proteins involved in the cellular antioxidant defense, including HMOX1, GPX1 and 4, and SOD2. Extensive amounts of reactive oxygen species (ROS) are known to be lethal, as well as to induce mitochondrial fission in response to increased mitochondrial depolarization[7]. In addition, while mitochondria constantly generate moderate amounts of ROS as a byproduct of respiration[7,29], certain OXPHOS impairments are well described as increasing ROS to detrimental quantities. To validate these proteomic results and investigate whether the potential increase in ROS following *Macrod1* knockdown is of mitochondrial origin, mitochondrial superoxide levels were measured via flow cytometry using MitoSOX™ in C2C12 cells 2 days after *Macrod1* knockdown; the respiratory chain complex I inhibitor rotenone was included as positive control. Indeed, loss of MacroD1 significantly increased mitochondrial superoxide production as measured by MitoSOX (Fig. 3E and Supplementary Fig. 5D), which could potentially compromise the electron transport chain. To further confirm that the mitochondrial fission phenotype observed here was a consequence of elevated mitochondrial ROS (mtROS) levels, C2C12 cells were treated with the antioxidant N-acetylcysteine (NAC) following *Macrod1* knockdown. Both mitochondrial fragmentation and cell death were rescued by NAC treatment (Fig. 3F–H and Supplementary Fig. 5E), suggesting that the observed increase in mtROS following *Macrod1* knockdown causes mitochondrial depolarization and fragmentation that culminates in apoptosis.

To determine if the enzymatic activity of MacroD1 is required for the mitochondrial fission phenotype, wildtype and ADP-ribose binding deficient MacroD1 proteins were overexpressed in C2C12 cells, and mitochondrial morphology analyzed via IF (Fig. 3I). Overexpression of mutant MacroD1, but not wildtype, induced

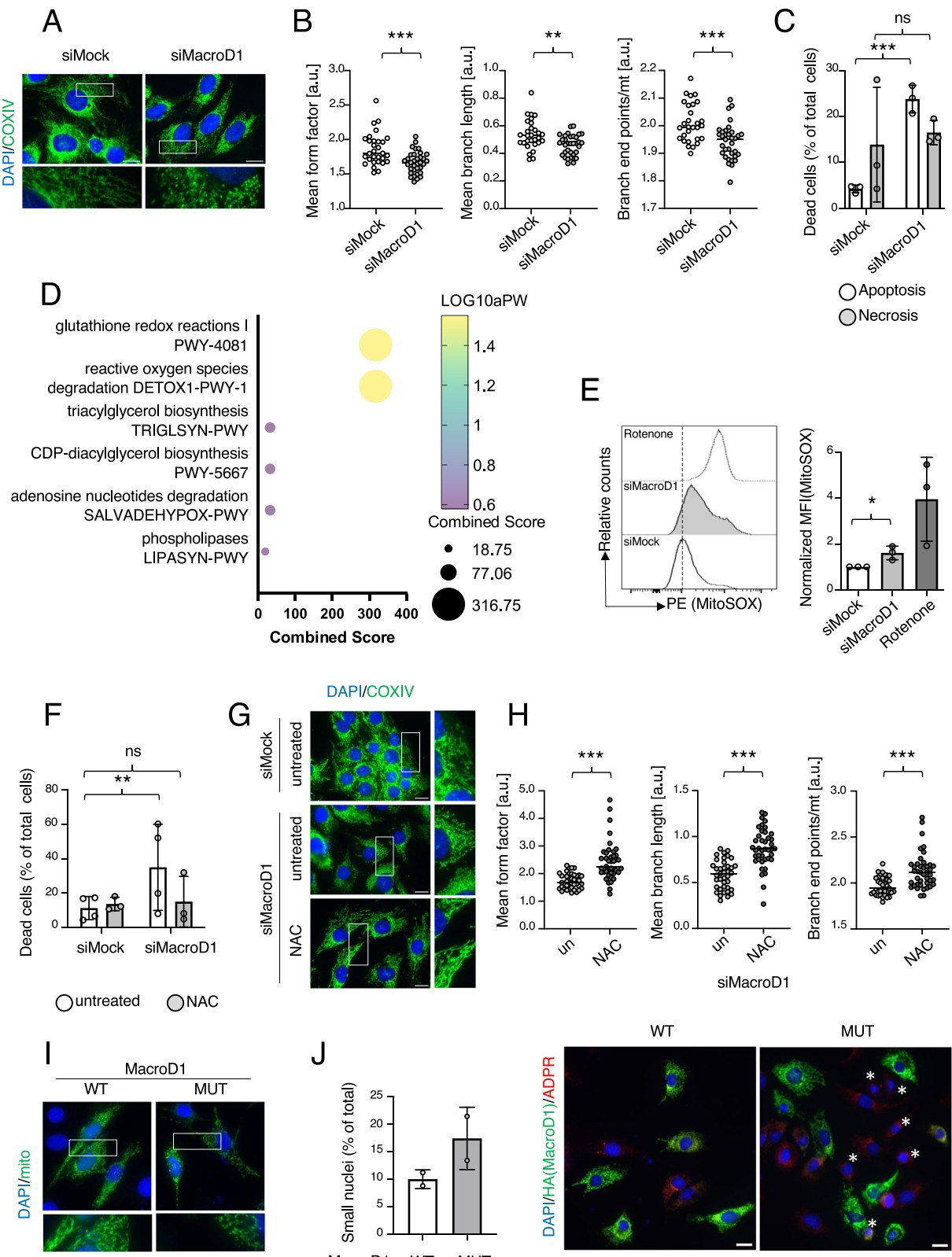

mitochondrial fission, which suggests that the morphological changes observed here are likely the result of a dominant negative effect. Moreover, comparable to what was observed following the extended knockdown of *Macrod1*, overexpression of the mutant also resulted in an increased proportion of smaller, more condensed nuclei in both C2C12 (Fig. 3J) and U2OS (Supplementary Fig. 5F). Together, these data suggest that the phenotypes observed

after MacroD1 knockdown were, indeed, dependent on the enzymatic activity of MacroD1.

## Loss of MacroD1 reduces the carbohydrate flux into the TCA cycle

To further understand the underlying cause for the increase in mitochondrial ROS production, we next focused on those proteins whose

**Fig. 3 | Knockdown of *Macrod1* leads to mtROS-mediated cell death and mitochondrial fragmentation.** *Macrod1* was knocked down for 2 days in C2C12 myoblasts, mitochondria were stained with an anti-CoxIV antibody (**A**) and morphological mitochondria features were extracted using the FIJI plugin Mitochondrial Analyzer (**B**). Data is shown as mean ± SD from n = 30 to 34 individual cells (technical replicates). Statistical analyses were performed using a two-tailed student t-test (*, $p < 0.05$; **, $p < 0.005$; ***, $p < 0.0005$). **C** *Macrod1* was knocked down for 3 days in C2C12 myoblasts and the proportion of apoptotic and necrotic cells was assessed via flow cytometry. Data is shown as mean ± SD from 3 biological replicates. Statistical analyses were performed using a two-tailed student t-test (*, $p < 0.05$; **, $p < 0.005$; ***, $p < 0.0005$). **D** Gene ontology analysis of all significantly upregulated proteins after 2 days of *Macrod1* knockdown in U2OS cells, as assessed by label free quantification-based whole-cell proteomics. **E** MacroD1 was knocked down in C2C12 myoblasts for 2 days, and mitochondrial superoxide production was analyzed by flow cytometry, using mitoSOX as a readout. Representative histograms are shown in the upper panel and bar graphs from three independent experiments on the lower one. Data is shown as mean ± SD from 3 biological replicates. Statistical analyses were performed using

a two-tailed student t-test (*, $p < 0.05$; **, $p < 0.005$; ***, $p < 0.0005$). **F–H** To address if antioxidant treatment can rescue cell death and mitochondrial morphology, 1 day after the knockdown of *Macrod1* cells were treated with NAC for 24 days, and cell death (**F**) and mitochondrial morphology (**G, H**) were analyzed via flow cytometry and immunofluorescence, respectively. For the cell death analysis, the data is shown as mean ± SD from 3 biological replicates. For mitochondrial quantifications, the data is shown as mean ± SD from n = 39–41 individual cells (technical replicates). For all statistical analysis, a a two-tailed student t-test was performed (*, $p < 0.05$; **, $p < 0.005$; ***, $p < 0.0005$). **I, J** MacroD1 wildtype (WT) or enzymatically inactive mutant (MUT) were overexpressed in C2C12 myoblasts, and mitochondrial morphology (**I**) and nuclear size (**J**) were analyzed via IF. **J** representative images are shown on the right, and condensed nuclei are marked with white asterisks. Data is shown as mean ± SD from 2 biological replicates with >100 single cells each (technical replicates). Statistical analyses were performed using a two-tailed student t-test (*, $p < 0.05$; **, $p < 0.005$; ***, $p < 0.0005$). Scale bars indicate 10 μm. Source data are provided as a Source Data file.

abundances decreased following knockdown of *Macrod1* (Supplementary Fig. 5B). Although there was no significant enrichment in GO terms, string analysis of all 239 down regulated proteins revealed an apparent enrichment of proteins involved in OXPHOS (Supplementary Fig. 6A, B). To determine if *Macrod1* knockdown reduces mitochondrial oxidative metabolism and alters carbon source utilization, the metabolite profile of C2C12 following *Macrod1* knockdown was analyzed via untargeted metabolomics (Supplementary Fig. 7A). In general, the levels of numerous metabolites were significantly altered following *Macrod1* knockdown compared to control cells. Intriguingly, three TCA cycle intermediates (α-ketoglutarate, fumarate, and malate) were significantly reduced following knockdown of *Macrod1* compared to the control samples. In contrast, glucose and several intermediates of the pentose phosphate pathway (PPP), including meso-erythritol, ribitol, and myo-inositol, were increased after the knockdown of *Macrod1* (Supplementary Fig. 7A). Furthermore, the amino acid serine, which can be synthesized from downstream metabolites of glucose, was also among the metabolites that were increased upon knockdown of *Macrod1*. These results suggest that *Macrod1* knockdown redirects glucose carbohydrate fluxes from mitochondria to the cytoplasm, potentially the PPP. To gain deeper insight into how MacroD1 might alter TCA metabolism, C2C12 cells were cultured in the presence of uniformly labelled stable isotope (U-$^{13}$C) glucose, and glucose breakdown was monitored via targeted metabolomics (Fig. 4A). 24 h after labelling, in both conditions (siMock and siMacroD1) most of the pyruvate and lactate was isotope labeled (Fig. 4B), suggesting that both metabolites were mainly generated from the breakdown of glucose and that *Macrod1* knockdown did not inhibit glycolysis. In line with the proteomics data and the results of the untargeted metabolomics approach, *Macrod1* knockdown significantly reduced the fraction of isotope labelled TCA cycle intermediates and associated metabolites (Fig. 4C). These findings indicate that MacroD1 is important for the metabolic flux from pyruvate into the TCA cycle. Importantly, while NAC treatment rescued the reduction in cell growth as observed before, it did not rescue the reduction in TCA cycle flux (Supplementary Fig. 7C, D), suggesting that the changes in mitochondrial metabolism are a direct consequence of MacroD1 loss that lead to the observed increase in ROS production and further cell death (Fig. 3). While changes in the incorporation of carbon atoms from glucose into most measured amino acids that are not directly connected to the TCA were not observed, there was also a significant increase in the amount of isotope-labeled serine (Supplementary Fig. 7B), again supporting an increase in the flux from glucose into the serine biosynthesis pathway.

In rapidly dividing cells, especially cancer cells, glutamine is avidly consumed and used for energy generation and as a carbon and

nitrogen source for biomass accumulation[29]. To determine if the reduced carbohydrate flux into the TCA cycle was specific to glucose breakdown products, targeted metabolome analyses were repeated using uniformly stable isotope-labelled (U-$^{13}$C) glutamine. Glutamine incorporation into the TCA cycle (i.e., into α-ketoglutarate, succinate, or malate) after MacroD1 knockdown was significantly increased compared to the control cells (Supplementary Fig. 7E), suggesting that these cells partially compensated for the reduced flux from pyruvate into the TCA cycle by increasing the flux from glutamate into the TCA cycle. Together, our proteomic and metabolomic data sets suggest that the loss of MacroD1 compromises OXPHOS and the incorporation of pyruvate into the TCA cycle. To finally validate the reduction in mitochondrial oxidative metabolism in a cell-based assay, a mitochondrial stress test was performed following *Macrod1* knockdown. As expected, the knockdown of *Macrod1* significantly decreased both basal and maximal respiration (Fig. 4D). Based on the observed reduction in OXPHOS, *Macrod1* knockdown should be detrimental under conditions that require cells to rely on mitochondrial metabolism. To address this, *Macrod1* knockdown C2C12 cells were subjected to glucose and glutamine starvation. While lack of glutamine slightly, but not significantly, increased cell death in both siMock and siMacroD1 cells, glucose starvation significantly and specifically increased cell death following *Macrod1* knockdown (Fig. 4E). Similarly, we also found that shMacroD1 C2C12 cells tended to be more sensitive to glucose starvation than their shMock counterparts (Supplementary Fig. 7F). Ultimately, taken together these data clearly demonstrate that MacroD1 is important for oxidative metabolism.

Phosphorylation and activation of Akt have been described as an important signaling event that correlates with glucose uptake in brown adipocytes and muscle[30]. Western blot and IF analyses were performed to determine if Akt phosphorylation was increased following *Macrod1* knockdown. Indeed, knockdown of *Macrod1* increased phosphorylated Akt (pAkt) levels both in Western blot and IF (Supplementary Fig. 7G and H), indicating that loss of MacroD1 activates Akt-mediated signaling. Together, these data suggest that the loss of MacroD1 rewires the cellular carbohydrate metabolism by increasing glucose dependency and redirecting metabolic fluxes from glucose towards the PPP and the serine biosynthesis pathway, resulting in carbohydrate metabolites missing for the TCA cycle.

### *Macrod1* knockdown promotes the pentose phosphate pathway to generate reducing equivalents and support the antioxidant response

In addition to the decrease in mitochondrial oxidative metabolism upon *Macrod1* knockdown, both metabolomics data sets hinted

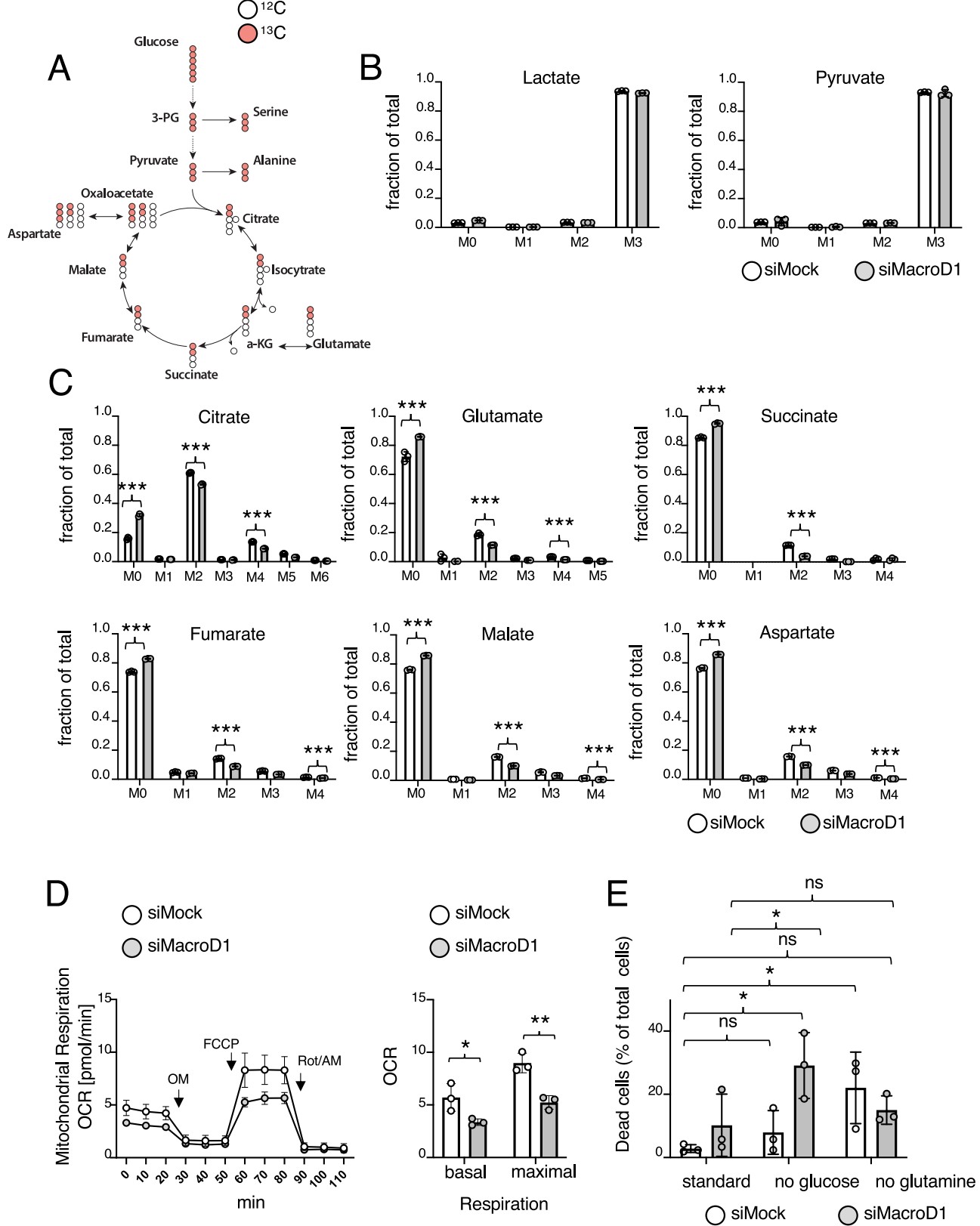

towards a potential redirection of glucose into the PPP (Supplementary Fig. 7A, B). The PPP plays an important role in the cellular response to ROS, as it generates NADPH which is required to reduce the antioxidant glutathione back from its oxidized form GSSG to GSH[31]. To test if, upon knockdown of *Macrod1* cells redirect glucose into the PPP to support an antioxidant response against the increase in mtROS

(Fig. 3E), we performed targeted metabolomics comparing a set of about 140 metabolites that are part of the central carbon metabolism (Fig. 5A). As expected, early glycolytic intermediates, such as fructose 1,6-phosphate and phosphoglyceric acid (Fig. 5B), as well as PPP intermediates, including ribose 5-phosphate and different nucleic acids (Fig. 5C) were among the most strongly increased metabolites

**Fig. 4 | Knockdown of *Macrod1* decreases the metabolic flux from glucose into the TCA cycle. A** C2C12 cells transfected with either siMacroD1 or a scrambled siRNA (siMock) were incubated in the presence of uniformly stable isotope labelled (U-$^{13}$C) glucose, and targeted metabolome analysis, monitoring the faith of glucose into different downstream metabolites were performed. Breakdown scheme of $^{13}$C glucose via glycolysis and TCA cycle; isotope labeled carbon atoms are depicted in red and non labelled ones in white. **B** Fractional labelling of lactate and pyruvate with M0, M1, etc. indicating the number of labeled carbon atoms, respectively. Data is shown as mean ± SD from 3 biological replicates. Statistical analyses were performed using a two-tailed student t-test (*, $p < 0.05$; **, $p < 0.005$; ***, $p < 0.0005$). **C** Fractional labelling of the indicated metabolites with M0, M1, etc. indicating the number of labeled carbon atoms, respectively. Data is shown as mean ± SD from 3 biological replicates. Statistical analyses were performed using a two-tailed student t-test (*, $p < 0.05$; **, $p < 0.005$; ***, $p < 0.0005$). **D** Two days after the knockdown of *Macrod1*, the oxygen consumption rate (OCR) of C2C12 cells was determined using a Seahorse analyzer, with addition of DMSO (as control) or oligomycin (OM), FCCP, or rotenone and antimycin (Rot/AM). Whole OCR curves are shown on the left, while comparisons between basal OCR (no treatment) and maximal OCR (post FCCP treatment) are shown on the right. Data is shown as mean ± SD from 3 biological replicates. Statistical analyses were performed using a two-tailed student t-test (*, $p < 0.05$; **, $p < 0.005$; ***, $p < 0.0005$). **E** Two days after knockdown of *Macrod1* C2C12 cells were subjected to either glucose or glutamine starvation, and cell viability was assessed via flow cytometry; Data is shown as mean ± SD from 3 biological replicates. Statistical analyses were performed using a two-tailed student t-test (*, $p < 0.05$; **, $p < 0.005$; ***, $p < 0.0005$). Source data are provided as a Source Data file.

upon knockdown of *Macrod1*. Notably, glutathione, as well as its precursor glycine, were also significantly increased upon knockdown of *Macrod1* (Fig. 5D), providing additional evidence that, as indicated by both the proteomic analysis and cell-based assays (Fig. 3D, E), lack of MacroD1 elevates intracellular ROS levels, thereby triggering an antioxidant defense response. To validate the hypothesis that absence of MacroD1 promotes the generation of reducing equivalents to support efficient ROS clearance, intracellular NADPH/NADP$^+$ levels were measured. In line with the hypothesis, the NADPH/NADP$^+$ ratio also significantly increased upon knockdown of *MACROD1* (Fig. 5E). This measurement represents only a snapshot analysis. While a time-course analysis would be interesting, *MARCOD1* knockdown already leads to increased cell death and the prolonged application of an additional stressor would lead to data with limited informative value.

To further confirm that the knockdown of *Macrod1* promotes the PPP to support NADPH generation without redirecting the carbon flux into lactate production, glucose uptake and lactate excretion were measured following knockdown of *Macrod1*. While there was no increase in total glucose uptake, there was a slight yet significant decrease in lactate excretion upon knockdown of *Macrod1*, ultimately resulting in a net reduction in aerobic glycolysis (Fig. 5F–H). In summary, these data suggest that cells respond to the increase in mtROS following loss of MacroD1 by redirecting their glucose flux into the PPP to support detoxification.

### MacroD1 regulates whole-cell NADH/NAD$^+$ ratios
Since MacroD1 was described to hydrolyze protein ADP-ribosylation in vitro[12], we investigated the effect MacroD1 had on mitochondrial ADP-ribosylation via IF using an antibody that detects ADP-ribose. In line with the idea that MacroD1 is an eraser of ADP-ribosylation, the knockdown of *Macrod1* resulted in an increase in mitochondria-associated signals (Supplementary Fig. 8A). To confirm that the observed increase in mitochondrial ADP-ribosylation was indeed the result of the loss of MacroD1, cells were genetically complemented with an siRNA-resistant MacroD1 construct after knockdown. Overexpression of this wildtype MacroD1 construct dampened the observed increase in mitochondrial ADP-ribosylation (Supplementary Fig. 8A), suggesting that MacroD1 functions as a mitochondrial ADP-ribosylhydrolase in vivo. To further elucidate the MacroD1-dependent mitochondrial ADP-ribosylome, MacroD1$^{-/-}$ and wildtype muscle tissues were lysed and analyzed via mass spectrometry using our well-established ADP-ribosylome workflow[32]. Surprisingly, we identified fewer ADP-ribosylated peptides and proteins in MacroD1$^{-/-}$ tissues compared to wildtype (Supplementary Fig. 8B, C, Supplementary Data 2). When taken together with the in vitro data presented above, these in vivo analyses suggest that MacroD1$^{-/-}$ animals have adapted to loss of MacroD1. The marginal effect that loss of MacroD1 had on protein ADP-ribosylation prompted us to analyze subcellular NAD$^+$ levels with previously described sensors[33]. Intriguingly, *MACROD1* knockdown significantly reduced mitochondrial, cytoplasmic, and

nuclear NAD$^+$ levels, comparable to the NAMPT inhibitor FK866, a known reducer of overall cellular NAD$^+$ levels that was used as positive control (Supplementary Fig. 8D). This decrease in NAD$^+$ very likely results from the observed increase in the PPP flux and the ratio of NADPH/NADP$^+$ described above (Fig. 5E). The ADP-ribosylated mitochondrial proteins identified here in the muscle of wildtype and MacroD1$^{-/-}$ animals were very similar to those previously described[34,35]. Cross-referencing the mitochondrial ADP-ribosylome with the recently published MacroD1 interactome[22] revealed that 4 of the ADP-ribosylated proteins identified here (SLC25A12, GLUD1, MDH2, and ATP5B) interacted directly with MacroD1. To gain further insights into the molecular networks regulated by MacroD1, we combined the mitochondrial ADP-ribosylome and MacroD1 interactome data and performed STRING analysis (Supplementary Fig. 8E). Very interestingly, many of the ADP-ribosylated proteins (highlighted in red) were found to be highly connected to proteins that interact with MacroD1 (highlighted in grey), likely because they function within the same protein complexes. Intriguingly, many TCA cycle enzymes and proteins/protein subunits involved in OXPHOS were either directly ADP-ribosylated or interacted with MacroD1 (Supplementary Fig. 8E, circled groups, and schematic localization in Supplementary Fig. 8F). This data further supports the conclusion that MacroD1 plays an important role in OXPHOS regulation.

### MacroD1 induces proteomic changes in muscle comparable to the ones observed in cell cultures
To strengthen the in vitro observations made in C2C12 cells, we also defined the functionally relevant changes on the whole cell proteome level in MacroD1$^{-/-}$ and wildtype skeletal muscle lysates ($n = 4$/group). In total, ~1200 quantifiable proteins were identified, and comparative analyses of the abundances of the proteins identified in MacroD1$^{-/-}$ and wildtype samples led to the identification of 94 proteins that were significantly differentially regulated (≥1.5 fold change, p-value ≤ 0.05; Fig. 6A, Supplementary Data 3). Of these 94 proteins, 35 were significantly upregulated and 59 were considerably downregulated in MacroD1$^{-/-}$ mice. To extract the mitochondrial proteins, we cross-referenced these differentially regulated proteins with the Mouse MitoCarta2.0 inventory[36]. Interestingly, 34% of the MacroD1 downregulated proteins identified here localize to mitochondria (Fig. 6B, nodes surrounded in black). STRING analysis of all down-regulated proteins led to the identification of a prominent cluster of genes involved in the TCA cycle (Fig. 6B, nodes highlighted in blue). Taken together, these data show that loss of MacroD1 results in a proteomic shift in muscle tissue in vivo.

### MacroD1 knockout mice strongly depend on glucose
Based on the comparable proteomics changes and the metabolic analyses, we reasoned that MacroD1$^{-/-}$ mice should also be more dependent on glucose than wildtype animals. We investigated this

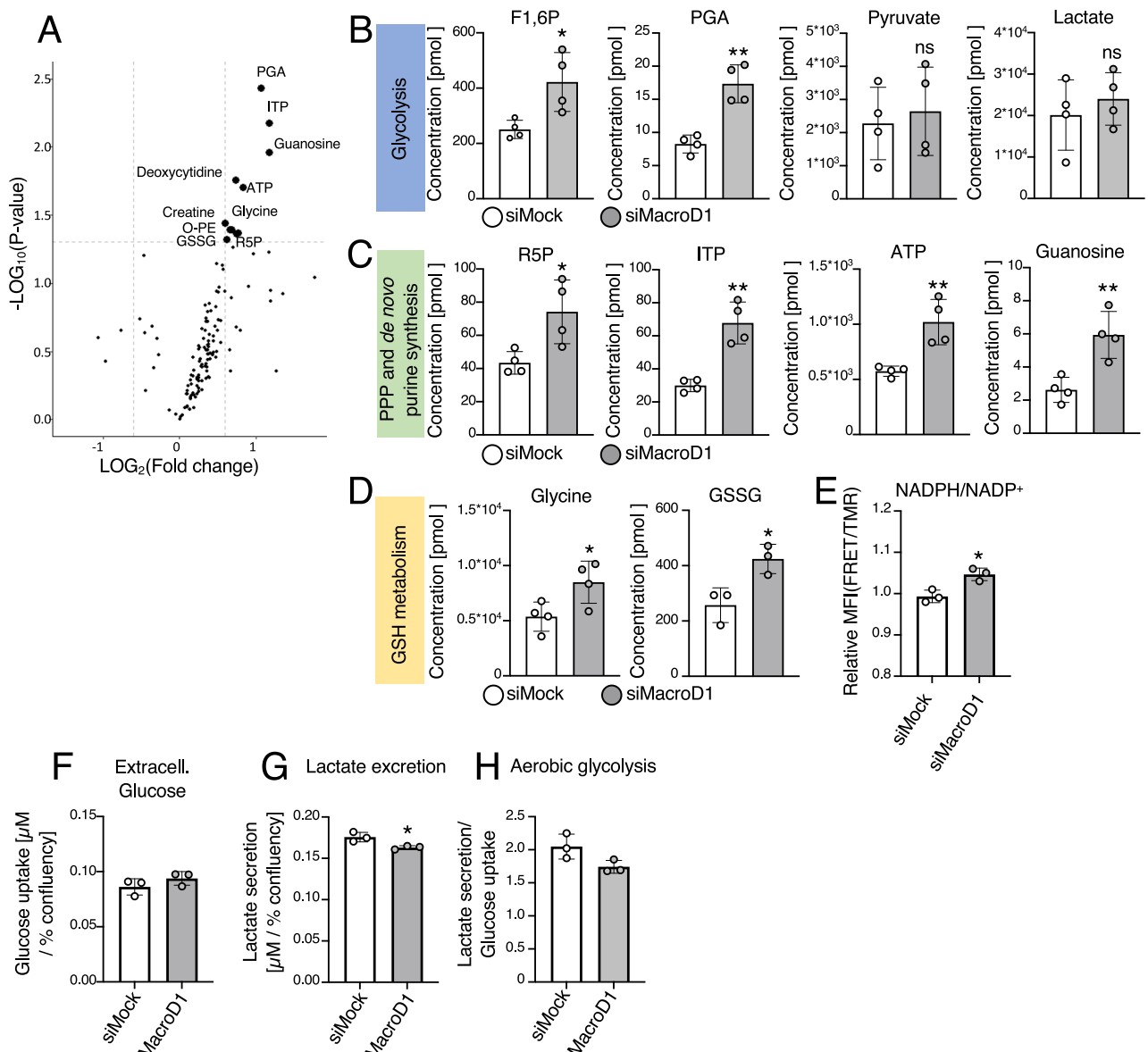

**Fig. 5 | Knockdown of *Macrod1* increases the production of antioxidants by promoting the metabolic flux into the pentose phosphate pathway. A** *Macrod1* was knocked down for two days in C2C12 cells, and metabolic changes were assessed by performing targeted metabolomics. Differentially detected metabolites are highlighted in bold; phosphoglyceric acid (PGA), inosine triphosphate (ITP), adenosine triphosphate (ATP), fructose-1,6-phosphate (F1,6 P), oxidized glutathione (GSSG), ribose-5-phosphate (R5P), O-phosphatidyl-ethanolamine (O-PE). **B–D** Bar graphs showing the changes of significantly altered metabolites from glycolysis (**B**), pentose phosphate pathway (PPP) and purine synthesis (**C**), and glutathione (GSH) metabolism (**D**); fructose-1,6-phosphate (F1,6 P), phosphoglyceric acid (PGA), ribose-5-phosphate (R5P), inosine triphosphate (ITP), adenosine triphosphate (ATP), oxidized glutathione (GSSG). Data is shown as mean ± SD from 4 technical replicates. Statistical analyses were performed using a two-tailed student t test (*, $p < 0.05$; **, $p < 0.005$; ***, $p < 0.0005$). **E** *Macrod1* was knocked down in stable U2OS Flp-In™ T-Rex™ cells expressing inducible cytoplasmic NADPH/NADP+ sensors, and NADPH/NADP+ ratios were assessed by flow cytometry. Data is shown as mean ± SD from 3 biological replicates. Statistical analyses were performed using a two-tailed student t-test (*, $p < 0.05$; **, $p < 0.005$; ***, $p < 0.0005$). **F–H** Glucose (**F**) and lactate (**G**) levels were measured in the medium of MacroD1 knockdown and control C2C12 cells and used to calculate the level of aerobic glycolysis (**H**). Data is shown as mean ± SD from 3 biological replicates. Statistical analyses were performed using a two-tailed student t-test (*, $p < 0.05$; **, $p < 0.005$; ***, $p < 0.0005$). Source data are provided as a Source Data file.

hypothesis using a glucose tolerance test. Mice were starved for 16 h and then injected with glucose, and blood glucose levels were monitored at various time points after injection. Blood glucose clearance rates were significantly faster in MacroD1⁻/⁻ mice than in wildtype animals, which is in line with the increased phosphorylation levels of AKT[30] (Fig. 6C). To investigate if this increase in glucose uptake was coupled with increased insulin sensitivity, mice were again starved overnight and subsequently injected with insulin. Following this procedure, 4 out of 5 female MacroD1⁻/⁻ mice immediately became

hypoglycemic, dictating the termination of the experiment. The fasting period was shortened to 4 h to avoid extreme hypoglycemia, and the experiment was repeated. Blood glucose levels of MacroD1⁻/⁻ animals dropped in response to insulin faster compared to wildtype animals (Fig. 6D). Moreover, blood glucose homeostasis was delayed in MacroD1⁻/⁻ animals, which suggests that loss of MacroD1 increases insulin sensitivity. Taken together, these data confirm that loss of MacroD1 results in a metabolic shift in cell culture and in vivo towards a higher dependency on glucose.

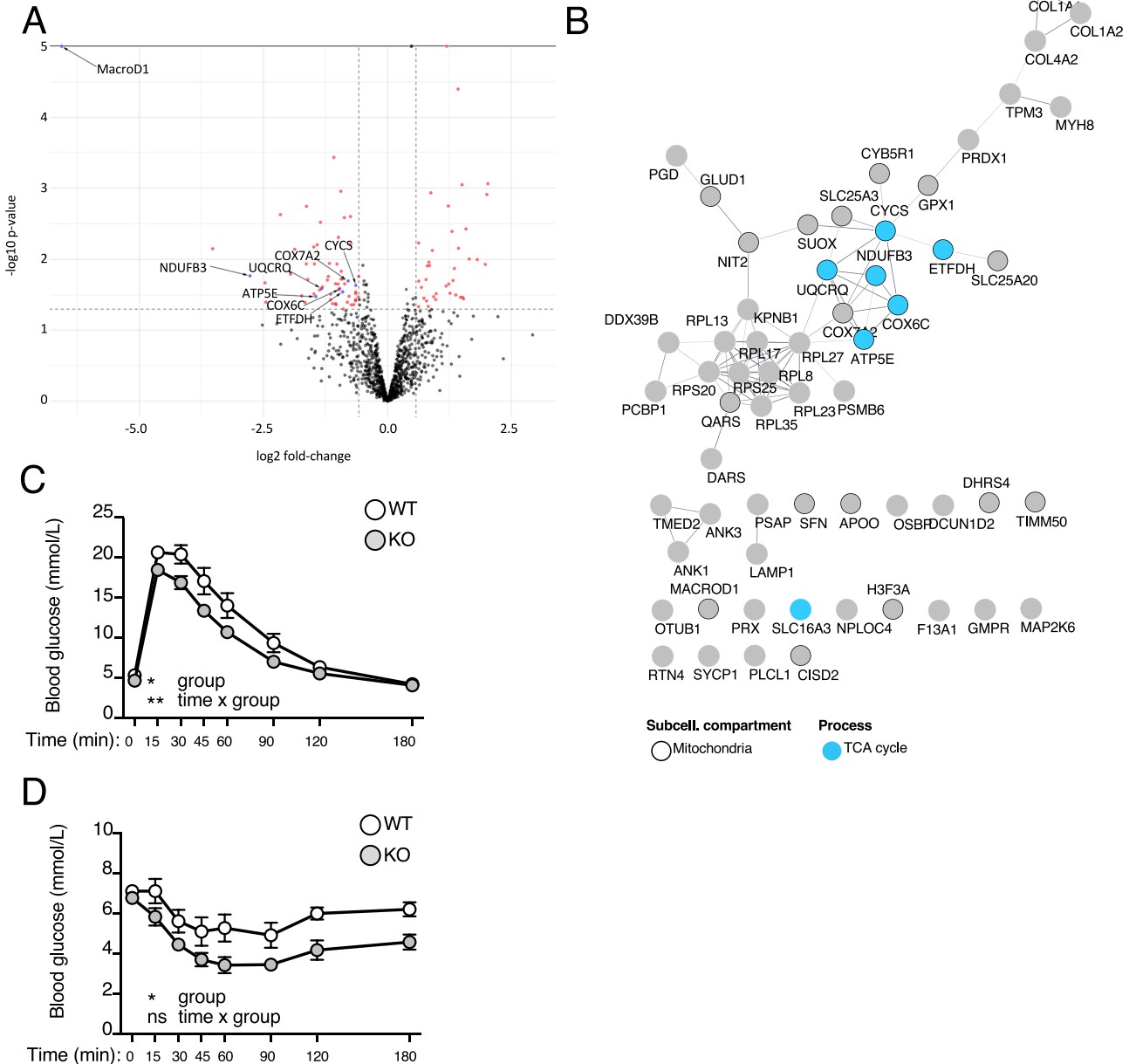

**Fig. 6 | Loss of MacroD1 alters the abundance of metabolic proteins and increases glucose consumption and insulin sensitivity. A** The whole proteome from the hind limb muscle of MacroD1$^{-/-}$ and wildtype mice was analyzed via mass spectrometry. Significantly altered proteins are highlighted in red (any significantly changed protein) or blue (highlighted changed proteins). **B** STRING analysis of significantly downregulated proteins. **C**, **D** MacroD1$^{-/-}$ (KO) and wildtype (WT) mice were starved for 16 h (**C**) or 4 h (**D**), and glucose (**C**) and insulin tolerance tests (**D**) were performed. Data is shown as mean ± SD from $n = 6$–8 individual mice. For statistical analysis of **C**, **D**, a two-way ANOVA with Geisser-Greenhouse correction as performed (*, $p < 0.05$; **$p < 0.005$; ***$p < 0.0005$) on data from $n = 14$–15 (**C**) and 6–8 (**D**) individual mice. Source data are provided as a Source Data file.

## Discussion

In this study, we identified MacroD1 to be crucial for mitochondrial integrity and function. In-depth proteomic and metabolomic analysis of siMacroD1 cells revealed that MacroD1 deficiency decreases the metabolite flux from glucose into the TCA cycle and redirects it into the PPP. Consequently, both siMacroD1 cells and MacroD1$^{-/-}$ mice showed an increased dependence on glycolysis and higher susceptibility towards glucose starvation.

MacroD1 has been described to hydrolyze protein- and nucleic acid-bound MAR modifications[12,13], as well as OAADPR, which is the by-product of sirtuin-catalyzed de-acetylation[14]. This is particularly intriguing since we recently described mitochondrial ADP-ribosylation[35],

and identified respiratory chain components like MDH1 and 2, ATP5a, or PDH to be ADP-ribosylated in cells and mouse organs[34,35,37]. Interestingly, a comparison of the recently published MacroD1 interactome[22] with our data revealed that many proteins/protein subunits that were found to be ADP-ribosylated either interact directly with MacroD1 or localize within proximity to components of the TCA cycle, complex I of the respiratory chain or the ATP-Synthase. In this regard, it is interesting to speculate that ADP-ribosylation might regulate the activity of the ATP-Synthase or other respiratory chain complexes and, thus, indirectly, the formation of mtROS[38]. Alternatively, loss of MacroD1 might increase O-acetyl-ADP-ribose, which could indirectly affect the ADP-ribosylation of the ATP-Synthase and

mtROS production. We also observed a reduction of NAD$^+$, together with an increase in NADPH/NADP$^+$ ratio in all subcellular compartments measured (mitochondria, cytoplasm, and nucleus), indicating that MacroD1 not only influences mitochondrial ADP-ribosylation but also TCA cycle and OXPHOS capacity and thereby the NADH/NAD$^+$ ratios in the cell. Mitochondria and the cytoplasm have distinct requirements for NAD$^+$, and proper compartmentalization of redox equivalents is crucial for maintaining cellular redox homeostasis and survival in response to environmental stressors[3–5]. The reduced respiratory capacity of mitochondria following loss of MacroD1 likely affects the compartmentalization of reducing equivalents through both the malate–aspartate shuttle and the citrate–malate shuttle[39,40]. Intriguingly, the 2-oxoglutarate malate shuttle was not only found to be ADP-ribosylated but also to interact with MacroD1[22].

Mechanistically, our data suggest that the knockdown of MacroD1 impairs respiratory chain function, which consequently leads to increased mtROS production. While mild doses of mtROS are beneficial and even required for proper communication between mitochondria and the nucleus, extensive and prolonged production of mitochondrial superoxide has been shown to have detrimental consequences for mitochondrial membrane polarization, function, and cell survival[41]. In line with this, we found that treatment with NAC (a ROS scavenger) was sufficient to rescue the *Macrod1* knockdown-induced mitochondrial fission phenotype and cell death. Although comparable experiments are challenging in animals, the TEM analysis of MacroD1 knockdown cells and muscles from MacroD1$^{-/-}$ animals revealed very similar morphological changes in mitochondria (i.e., reduced mitochondrial cristae and network formation). The same molecular mechanisms likely induced the observed changes and most probably affected the animals' maximal exercise capacity.

The proteomic changes observed following the MacroD1 knockdown comprised an increase in proteins involved in the cellular antioxidant response and a decrease in proteins involved in acetyl-CoA metabolic processes, particularly pyruvate dehydrogenase (PDH). This is consistent with the observed changes in oxidative metabolism in mouse muscle. Moreover, the observed downregulation of proteins involved in nucleotide metabolism might provide the first evidence of why sustained knockdown of *Macrod1* might result in cell death. Concurrent with the proteomic changes observed here, in-depth metabolomic analyses of cells following *Macrod1* knockdown revealed that MacroD1 deficiency rewires the metabolite flux of glucose and results in its redirection from the TCA cycle to the PPP and serine biosynthesis pathway. Both pathways have been described as important components of the antioxidative system, as they support NADPH production, which functions as an indispensable cofactor for both glutathione reductase and thioredoxin reductases[42,43]. Together, these findings suggest that redirecting the carbohydrate fluxes in this manner could be an important response mechanism that allows the cell to cope with mtROS increases caused by compromised mitochondrial functions following a loss of MacroD1.

In support of these hypotheses, MacroD1 was previously shown to be transcriptionally co-regulated with various enzymes involved in the TCA cycle and the respiratory chain[44]. Here, loss of MacroD1 resulted in a decrease in the incorporation of pyruvate but an increase in the incorporation of glutamate into the TCA cycle. Consequently, loss of MacroD1 partitioned the TCA cycle in two fractions, in which only the flow from α-ketoglutarate to oxaloacetate was somewhat comparable to wildtype cells. Thus, given that both *Macrod1* knockdown cells and MacroD1$^{-/-}$ mice were found to be more dependent on glucose and more susceptible to glucose starvation, the increase in glutamine incorporation was likely not sufficient to compensate for the overall lack of carbohydrate incorporation from glucose into the TCA cycle. This line of reasoning aligns with recent studies that have suggested that MacroD1 might play a role in regulating adipogenesis as well as

insulin receptor signaling in L1-3T3 and MIN6 cells[24,45,46]. Indeed, *Macrod1* knockdown in pre-adipocytes was shown to promote insulin receptor signaling and glucose uptake and increase PPARγ expression on both mRNA and protein levels[24].

Given the strong impact that MacroD1 loss had on cell metabolism and thus the distribution of important small molecule cofactors such as NAD$^+$ or acetyl-CoA inside the cell, the MacroD1 level could potentially directly contribute to gene regulation and cell identity[47]. Indeed, our proteomics data revealed that many proteins involved in genome architecture are differentially regulated upon knockdown of *Macrod1*. In line with that, knockdown of *Macrod1* or over-expression of an enzymatically dead mutant resulted in nuclear condensation. Given our data, it would be interesting to investigate if and how MacroD1 contributes to regulating nuclear dynamics and how this might relate to some of the MacroD1 dependent phenotypes observed by us and others. Although the in vivo part of our study focused on muscle function, we also observed an increase in the fat mass of MacroD1$^{-/-}$ animals. These results are in line with a previous study suggesting that MacroD1 is a negative regulator of adipogenesis[24]. The impact of the loss of MacroD1 on other metabolically active tissues (e.g., liver, brain) requires further investigation. Nevertheless, the necropsy performed here did not reveal any macroscopic changes in chow-fed mice up to 8 weeks old. However, gene expression studies of BAT isolated from MacroD1$^{-/-}$ animals showed comparably reduced expression of genes involved in metabolic regulation. This suggests that the observed influence of MacroD1 on mitochondrial structure and function might be a general consequence of loss of MacroD1 and not a cell-type-specific phenomenon. Our findings strongly indicate that MacroD1 functions as a positive regulator of oxidative metabolism. Intriguingly, MacroD1 expression levels and mutation within the gene have been shown to positively correlate with the aggressiveness and invasiveness of various types of cancer[48,49]. This is particularly interesting given that cancer cell metabolism is commonly associated with increased anaerobic glycolysis over OXPHOS[50]. Generally, MacroD1 expression levels positively correlate with proliferation rates and tumor invasiveness[48,49,51–53]. Several follow-up studies linked the effect that MacroD1 had on tumor growth and progression with estrogen- and androgen-receptor signaling[54–57]. Based on our findings and the positive correlation between MacroD1 and cancer invasiveness, MacroD1 might be an attractive new target for cancer therapy. Moreover, the ADP-ribosylation status of specific mitochondrial proteins could be a diagnostic tool to predict cancer aggressiveness, progression, and metastatic potential.

## Methods
### Animals and animal experiments
The here reported study complies with all relevant ethical regulations. All animal experiments were carried out following the Swiss and European Union ethical guidelines and have been approved by the local experimental committee of the Canton of Zurich under licenses 276/2014, 207/2015, and 035/2016. Heterozygous C57BL/6N MacroD1$^{-/-}$ animals (*M. musculus*) carrying a CRISPR/Cas9-mediated deletion of the last 7 exons (exons 5-11) of MacroD1 were purchased from the Wellcome Trust Sanger Institute (Hinxton, Cambridge) and crossed amongst each other to generate homozygous MacroD1$^{-/-}$ whole body knockout animals and the respective wildtype MacroD1$^{+/+}$ controls. All animals were housed under pathogen-free conditions at the University of Zurich. Mice were housed on a 12h-12h light-dark cycle (6:00/18:00) with *ad libitum* access to water and standard rodent chow (Kliba Nafag maintenance extrudate for mice and rats; product no. 3436). Mice were weighed weekly to monitor their weight gain. Whole mouse magnetic resonance imaging (MRI) measurements were performed using the EcoMRI™. Mice were immobilized in a Plexiglas tube that was subsequently inserted into the machine.

## Food intake measurements

To investigate food and water intake and excretion, mice were housed individually for 3 subsequent days in metabolic cages, and their weight, food intake, feces, water uptake, and urine were measured daily. To this end, food trays were weighed every day prior to and after refilling, feces were collected and weighed, and urine was measured using a pipette. Before the experiment, mice were acclimatized to the metabolic cages for several days by allowing them to inspect the cage for an increasing amount of hours each day.

## Exercise performance

Animals were forced to run on a motorized, speed-controlled metabolic treadmill modulator equipped with electrical stimulus grids and an air-tight enclosure around the running belt until exhaustion. Mice were allowed to acclimatize for 15 minutes before running for 5 min at the lowest speed and without inclination. After that, the pace was increased every min by 1 m/min, and the inclination was increased every 2 min by 5° until a maximal inclination of 20° was reached. Running time was stopped when a mouse (i) stayed on the electric grid for more than 3 consecutive sec; (ii) jumped off the electric grid 3 times without re-engaging in the running; (iii) stayed on the electric grid for 2 consecutive sec for the third time.

## Intraperitoneal glucose and insulin tolerance test (IPGTT and IPITT)

For the IPGTT mice were starved for 16 h overnight with *ad libitum* access to water: Following the intraperitoneal (ip) injection of 2 mg of glucose/g of body weight, blood glucose levels were measured after 15, 30, 45, 60, 90, 120 and 180 min by extracting a blood drop from the tail vein. For the IPITT, animals were starved for 4 h, and 0.75 mU insulin/g body weight was injected ip, before blood glucose levels were monitored at various time points.

## Phenotyping of MacroD1$^{-/-}$ mice and histological, immunohistochemical, and ultrastructural examinations

For phenotyping, 3 female 6-week-old MacroD1$^{-/-}$ and 2 female 6-week-old wildtype MacroD1$^{+/+}$ mice were sacrificed and subjected to a complete post-mortem examination and histological examination of a wide range of organs/tissues (brain, heart, trachea and lungs, tongue, salivary glands, esophagus, stomach, duodenum, jejunum, ileum, caecum, colon, liver with gallbladder, pancreas, kidneys, urinary bladder, spleen, mandibular and mesenteric lymph nodes, thymus, gut-associated lymphatic tissue, bone marrow (femur, sternum), ovaries, uterus, vagina, skeletal muscle (*M. quadriceps*, *M. gastrocnemius*, diaphragm), femoral bone, head with nasal cavity and teeth, spinal cord, pituitary gland, thyroid and parathyroid glands, adrenal glands, interscapular brown adipose tissue, skin and mammary gland). Tissue samples were fixed in 4% paraformaldehyde (PFA), trimmed, and routinely paraffin wax embedded. Sections (3-4 μm) were prepared and routinely stained with hematoxylin-eosin (HE). Additional samples were collected from the *M. quadriceps femoris*, the diaphragm, and the heart for transmission electron microscopy (TEM). These were fixed in 2.5% glutaraldehyde (EMS) buffered in 0.1 M Na-P-buffer overnight, washed 3 times in 0.1 M buffer, post-fixed in 1% osmium tetroxide (Sigma-Aldrich) and dehydrated in ascending concentrations of ethanol followed by propylene oxide and infiltration in 30% and 50% Epon (Sigma-Aldrich). At least one semithin section (0.9 μm thick) was prepared and stained with toluidine blue from each tissue. Representative areas were selected and trimmed. Ultrathin sections (90 nm thick) were prepared, contrasted with lead citrate (Merck) and uranyl acetate (Merck), and viewed under a Phillips CM10 TEM, operating with Gatan Orius Sc1000 digital camera, using a Gatan Microscopical Suite, Digital Micrograph, Version 230.540. To measure the lengths of the Z lines, ultrathin sections with well orientated myofibres were selected. With distinct cell borders, the length of Z-lines was determined, with indistinct cell borders, the distance from Z- to Z-line was measured ($n = 5-6$ per section). To determine the mitochondrial load, mitochondria were manually counted in sections derived from 4 knockout and 4 wildtype animals each.

For further examinations, a larger cohort of MacroD1$^{-/-}$ and MacroD1$^{+/+}$ mice (5 males and 5 females of each genotype) was sacrificed via $CO_2$ inhalation at the age of 6–7 weeks. The right *M. quadriceps femoris*, *M. gastrocnemius*, *M. soleus*, and *M. triceps brachii* and the heart were dissected, and samples of each frozen onto an OCT-covered (CellPath, Powys, UK) thin cork plate in isopentane pre-cooled in liquid nitrogen. Skeletal muscle samples were oriented with the fibers perpendicular to the plate, and the heart samples were oriented with the longitudinal axis of the heart perpendicular to the plate to obtain cross-sections. Additional samples (appr. 2 mm$^3$) from the muscles and heart were collected, fixed in glutaraldehyde, and processed for TEM as described above.

From the cryoblocks, consecutive sections (8 μm) were prepared and routinely stained with hematoxylin and eosin (HE), for succinic dehydrogenase (SDH) activity and by immunohistochemistry[58]. The SDH histochemical stain identifies enzymatic activity by incubating fresh frozen tissue sections in 0.2 M phosphate buffer (pH 7.6; Merk, Germany) containing 0.2 M succinic acid (S2378, Merk, Germany) and 10 mg nitro blue tetrazolium (N6876, Merk, Germany) at 37 °C for 3 h. Stained sections are washed, treated with graded acetone, and mounted in an aqueous medium for evaluation.

Immunohistochemistry served to detect type I myofibers. Briefly, endogenous peroxidases were blocked by incubation with hydrogen peroxide (Agilent, Santa Clara, CA, USA) solution for 10 min. Slides were then incubated for 1 h with mouse anti-myosin heavy chain 7 (MyHC 7, clone NOQ7.5.4D for type 1 fibers[59]) diluted 1:6,000 in REAL Antibody Diluent (Agilent, Santa Clara, CA, USA), followed by the MACH4 detection system (Biocare Medica, Concord, CA, USA) and DAB substrate buffer (Agilent, Santa Clara, CA, USA) as detection chromogen. The staining was performed in an AutostainerLink48 (Agilent, Santa Clara, CA, USA). Sections were washed with phosphate-buffered saline (pH 8) between each incubation step. Finally, sections were counterstained with hematoxylin for 40 sec and mounted. Sections from the *M. quadriceps femoris* of a C57BL/6N mouse served as positive controls, and sections incubated without the primary antibody served as negative controls.

The computer program Visiopharm (Version 2023.01.3.14018; Visiopharm, Hoersholm, Denmark) served for the morphometric analyses, i.e., to measure the lesser diameter of myofibres in a 1 mm$^2$ area of the HE-stained cryosections of all muscle samples, to measure the mean staining intensity in whole sections of the *M. triceps brachii* stained for SDH activity, and to count type I (positive) and type II (negative) fibers in the *M. soleus* stained for MyHC 7.

## Cell culture

All cell lines used for this study were initially purchased from ATCC or Millipore, grown under sterile conditions (humidified atmosphere, 5% $CO_2$, 37 °C), and routinely tested for mycoplasma infection. Human U2OS and HEK293T cells, as well as murine NIH-3T3 and HL-1 cells, were cultured in high glucose-containing Dulbecco's modified Eagle's medium (DMEM) supplemented with 5% penicillin/ streptomycin (P/S) and 10% (v/v) fetal calf serum (FCS). Murine C2C12 cells were grown in high glucose and pyruvate-containing DMEM, supplemented with 5% P/S and 20% (v/v) FCS, and maintained below confluency to avoid differentiation. Stable U2OS Flp-In T-Rex cells expressing inducible NAD$^+$ sensor constructs were cultured in high glucose-containing DMEM supplemented with 5% P/S and 10% (v/v) FCS in the presence of 100 μg/mL hygromycin and 15 μg/mL Blasticidin, and 200 ng/mL doxycycline was added overnight if sensor expression was required.

## Cell treatments

Unless otherwise stated, cells were treated with the following compounds at the indicated concentrations and for the stated amount of time: NAC (50 mM) for 24 h. For stable isotope labelling, the medium was exchanged by tracer medium where the substrate of interest, glucose or glutamine, was substituted by its fully $^{13}C$-labeled counterpart ($[^{13}C_6]$-glucose (25 mM) or $[^{13}C_5]$-glutamine (4 mM)) and cells were incubated for additional 36 h before metabolite extraction.

## Cell transfection and viral transduction

Transient overexpression of MacroD1, MacroD2, or Targ in HEK293T cells was achieved via calcium phosphate transfection. Roughly 12 h after transfection, the medium was removed and replaced with fresh DMEM. Experiments were started 2 days after transfection.

Lentiviruses were produced by co-transfecting HEK293T cells with viral constructs, packaging- and envelope plasmid in a ratio of 1:1:2 using calcium phosphate. In brief, $3*10^6$ HEK293T cells were seeded on a 10 cm dish the day before transfection. The next day, 10 μg of plasmid mixture was diluted in 500 μL $H_2O$, supplemented with 248 mM of $CaCl_2$. 500 μL 2x HBS (280 mM NaCl, 10 mM KCl, 1.5 mM $Na_2HPO_4$, 12 mM dextrose, and 50 mM HEPES, pH 7.5 in $H_2O$) were added dropwise and the whole transfection cocktail was well mixed. After 4 min of incubation at RT (the transfection crystals should roughly grow to the size of the cell's nuclei), the mix was added to the cells, and about 12 h after transfection, the medium was removed and replaced with fresh DMEM. The medium was collected two days after transfection, spun down at $400 \times g$, and filtered through a 0.45 μm mesh to eliminate residual cells. Approximately 40% of confluent U2OS cells seeded in a 6-well format were transduced with 2 mL of the virus-containing medium, supplemented with 5 mg/mL polybrene. After 24 h of recovery, cells were subjected to Blasticidin selection.

## siRNA transfection

siRNA-mediated knockdown of MacroD1 was performed via reverse transfection using Lipofectamine RNAi MAX according to the manufacturer's manual. In brief, cells were seeded into a 6 cm dish, at roughly 40% confluence, and 30 pmol siRNA were mixed with 5 μL lipofectamine in 500 μL serum-free OptiMEM and incubated for 20 min at room temperature (RT) before being added dropwise onto the cells minutes after seeding. 24 to 72 h after siRNA transfection, downstream experiments were performed. A scrambled siRNA was used as a control for each experiment. To knock down mouse *Macrod1* the following sequences were used (CAAGGATTTCATTAAGCTGAA) and (AAGTATAAGAAGGACAAGCAA) and for human *MACROD1* (CCCGAGGAGCCUAAUAAAGAUTT) and (CCCGGCCAAGTACGTCATCCA).

## MitoTracker™, TMRE, and MitoSOX™ staining

The cell-permeable fluorescent dye MitoTracker™ Green FM (Thermo Fisher) that stains active mitochondria was used to quantify absolute mitochondrial loads of cells by flow cytometry according to the manufacturer's manual. Cells were seeded in 6-well format, and 2 days after siRNA transfection, the cell culture medium was removed and replaced with 2 mL PBS per well containing 100 nM MitoTracker. Cells were stained for 30 min in the incubator before being washed 2× with PBS, harvested, and resuspended in 500 μL PBS. The dye's fluorescent intensity was recorded in life cells via flow cytometry.

The TMRE-mitochondrial membrane potential assay kit (Abcam) was used to assess relative mitochondrial membrane potentials in life cells via flow cytometry according to the manufacturer's manual. As for MitoTracker, cells were seeded and reverse-transfected with siRNA for 2 days. The medium was exchanged with 2 ml PBS per well containing 300 nM TMRE, and cells were stained for 30 min in the incubator before being washed 2× with PBS, harvested, and analyzed via flow cytometry.

The mitochondria-specific superoxide indicator MitoSOX ™ Red (Thermo Fisher) was used to quantify mitochondrial ROS production in life cells by flow cytometry according to the manufacturer's manual. After seeding and 2 days of siRNA transfection (as described for MitoTracker), the medium was exchanged with 2 mL of PBS containing 5 μM MitoSOX for 10 min in the incubator before being washed 2x with PBS, harvested, and analyzed via flow cytometry.

Fluorescence intensities of the respective dyes in life cells were analyzed using the LSR II Fortessa with the following laser and filter combination: 405 nm to acquire forward and sideward scatter, 488 (505LP, 530/30) to acquire the MitoTracker signal, and 561 (586/15) to acquire the TMRE and the MitoSOX™ signal. The data were analyzed using FlowJo.

## Immunofluorescence (IF)

For all IF experiments, cells were grown on 12 mm glass coverslips, treated when required, fixed in 4% paraformaldehyde for 15 min at RT, and permeabilized for 10 min at RT in PBS supplemented with 0.2% Triton-X100 (Sigma Aldrich). When nuclear pre-extraction was required, cells were incubated in an ice-cold permeabilization solution for 2 min before being fixed. Cells were blocked in filtered PBS supplemented with 0.1% Triton-X100 and 2% BSA for 1 h. The primary antibodies were diluted in the same buffer, and cells were incubated with the antibody solution overnight at 4 °C. The secondary antibodies were diluted in the standard blocking solution (0.1% Triton-X100 and 2% BSA in PBS) for 1 h at RT. After each antibody incubation, cells were washed 3 times with PBS. Following the final wash, cells were incubated with 0.1 μg/mL DAPI in PBS for 20 min at RT. After an additional PBS wash, the coverslips were briefly washed in distilled water and mounted on glass slides using 5.5 μL Mowiol solution per coverslip. The following antibodies were used for IF at the indicated dilutions: mouse anti-ATP5a (Abcam, clone 15H4C4, 1:250), mouse anti-HA (Biolegend Poly9023, 1:250), rabbit anti-COXIV (Abcam ab16056, 1:500), rabbit anti-ADPR (Hottiger laboratory, 1:500).

## Confocal microscopy

Confocal images were acquired on an automated CLSM – Leica SP8 upright confocal laser scanning microscope equipped with 4 solid state diode lasers (405, 488, 552, and 638 nm) using an HCX PL APO CS2 ×63 immersion oil objective. For all images, brightness and contrast were adjusted using FIJI. The same acquisition and image processing settings were used for all images within one experiment.

## Quantification of mitochondrial ADP-ribosylation

Quantification of mitochondrial ADP-ribosylation signals by IF was performed as previously described[60]. In brief, cells were prepared and stained as described above (see IF section), using the anti-ATP5a and the anti-ADPR antibodies. Images were taken on the Olympus ScanR screening system using the UPLSAPO ×20 objective (NA 0.9). Following the acquisition, images were analyzed using the Olympus ScanR Image Analysis Software version 3.0.1. After a dynamic background correction was applied, image segmentation was performed based on the DAPI signal in order to identify cell nuclei as individual objects. Further, mitochondria were identified as associated objects using similar intensity-based segmentation based on ATP5a co-staining, within an area spaced minimally 1.6 μm and maximally 26 μm from the nuclear periphery. Mean fluorescence intensities within the nuclear or mitochondrial masks were quantified per cell and are displayed as cell population bar graphs using GraphPad Prism 8.0.

## Analysis of mitochondrial morphology

After image acquisition, mitochondrial morphology was either assessed manually, or using the FIJI plugin Mitochondrial Analyzer[61]. For manual analysis, a minimum of 7 images per condition were used and mitochondrial morphology of every single cell per picture was

assessed manually in a double-blinded manner. Only after the analysis of a whole experiment, conditions were associated to the respective images. For computational analysis, all required plugins and updates were installed in FIJI according to the user manual (https://github.com/AhsenChaudhry/Mitochondria-Analyzer). Further, images with mitochondrial signal only (e.g., COXIV antibody signals) were opened in FIJI, thresholds were selected according to the magnification of the respective images and mitochondrial parameters were analyzed using the 2D analysis tool.

## Seahorse analysis

Cells were seeded in 96-well plates at 2000 cells/well for siMock and 2500 cells/well for siMacroD1. 8 h later siRNAi transfection was performed as described above (20 nM). 24 h after transfection, the seahorse cartridge was hydrated as indicated in the manual. 48 h after siRNAi transfection, before measurement, medium was exchanged for DMEM XF Base Medium pH 7.4 (Agilent 102353-100) containing glucose (10 mM), sodium pyruvate (1 mM), and l-glutamine (2 mM), Hepes (5 mM) and cells were incubated for 1 h at 37 °C. Measurements were carried out on a Seahorse XF96 (Agilent) with a MitoStress (Agilent, 103015-100) kit, following the manufacturer's instructions. Oligomycin, carbonyl cyanide 4-(trifluoromethoxy)phenylhydrazone, and a mix of Rotenone and Antimycin A were injected at desired time points at a final concentration of 1, 2, and $1 + 1 \mu M$, respectively.

## Quantification of Mitochondrial DNA Copy Number

Mitochondria DNA copy number was measured by the ratio of a mitochondrial gene, 16 s rRNA and a nuclear gene, HK2 as described[62]. Briefly, DNA was extracted from mouse organs using the High Pure PCR Template Preparation Kit (Roche, 11796828001) according to the manufacturer's instructions. qPCR was performed according to standard protocols using 25 ng of DNA per reaction and was done in triplicates on a QuantStudio 5 Real Time PCR system (Thermofisher) with KAPA SYBR Fast (KAPA Biotechnologies). The following primers were used:

| 16S rRNA mouse Fw | CCGCAAGGGAAAGATGAAAGAC |
| 16S rRNA mouse Rw | TCGTTTGGTTTCGGGGTTTC |
| HK2 mouse Fw | GCCAGCCTCTCCTGATTTTAGTGT |
| HK2 mouse Rw | GGGAACACAAAAGACCTCTTCTGG |

The relative mtDNA copy number was analyzed by the $2^{-\Delta\Delta Ct}$ method.

## Intracellular, compartmentalized NAD$^+$ and NADPH/NADP$^+$ measurements

Stable U2OS Flp-In cells, expressing inducible NAD$^+$ or NADPH/NADP$^+$ sensors[33] were seeded in 6-well format, and a reverse siRNA transfection was performed as described above. The day after transfection, cells were induced overnight with 200 ng/mL doxycycline. The day after induction, cells were labeled with 500 nM CP-TMR-C6-SMX and 500 nM SiR-Halo, again overnight. After labeling, cells were washed at least three times thoroughly with DMEM to remove any excess of the dyes and, if needed, subjected to a 1 μM Rotenone treatment for 1 h. Finally, cells were trypsinized and collected in 500 mL PBS. The FRET ratio, as a proxy for NAD$^+$ levels, was determined via flow cytometry using the LRS II Fortessa. The following laser and filter combination was used: 405 nm to acquire forward- and sideward scatter, 561 (586/15) to acquire the donor signal (CP-TMR-C6-SMX), 561 (635 LP, 670/30) to acquire the FRET signal and 640 (670/14) to acquire the acceptor (SiR-Halo) only. The data were analyzed using FlowJo. The FRET ratio was calculated by dividing the FRET signal by the donor signal.

## Western blotting

For WB analysis, proteins were separated via SDS-page on a 12% SDS-polyacrylamide gel at 120 V. A wet-transfer into a PVDF membrane was performed at 30 V over-night and membranes were blocked with 5% milk in TBS-T for 1 h at RT. Primary antibodies were diluted in 1 % milk in TBST and incubated at 4 °C over-night. After 3 washes, the secondary antibody, diluted in TBST, was incubated for 1 h at RT. After another 3 washes, specific proteins/bands were visualized with the Odyssey infrared imaging system (LI-COR).

## Statistical analysis

Normally distributed quantitative data were compared using the Student's t-test for two groups or one-way ANOVA followed by Tukey's multiple comparisons for more than two groups. For data sets comparing two groups over time, a two-way Anova with Geisser-Greenhouse correction was used. For non-normally distributed quantitative data, the Mann–Whitney U test was used for comparisons between two groups, while the Kruskal-Wallis test followed by Dunn's multiple comparisons was applied when more than two groups were involved. Categorical variables were compared using the Chi-squared test. A p-value of 5% or less was considered statistically significant.

## Whole-cell/tissue lysate preparation, trypsin digestion, and ADP-ribosylated peptide enrichment

Cells and tissues were lysed in lysis buffer (6 M Gnd-HCl, 50 mM Tris pH 8.0), sonicated, and stored at −80 °C until LC-MS/MS analysis. Protein disulfide bridges were reduced with 5 mM Tris(2-carboxyethyl) phosphine (TCEP) and alkylated with 10 mM 2-Chloroacetamide (CAA) in the dark at 37 °C for 1 h. For whole proteome analyses, 50 μg of proteins were prepared for digestion using the filter-aided sample preparation (FASP) methodology[63] and digested with Sequencing Grade Trypsin (1:25; Promega) overnight at 37 °C. The samples were then acidified with TFA, and salts were removed using ZipTip C$_{18}$ pipette tips (Millipore Corp.). The peptides were eluted with 15 μL of 60% ACN, 0.1% TFA, dried to completion, and then re-dissolved in 3% ACN, 0.1% formic acid to a final peptide concentration of 0.5 μg/μL.

For ADP-ribosylome analyses, 15 mg of proteins were diluted 1:30 with PARG buffer[32] and digested with modified Porcine Trypsin (1:25; Sigma) overnight at 37 °C. ADP-ribosylated Peptide enrichments were carried out as described[32] with the following protocol modifications. Following PARG-mediated PAR-to-MAR reduction and SepPak clean-up, the peptides were enriched using Af1521-WT (0.5 mL beads/15 mg lysate,[32]) and eAF1521 (1.0 mL beads/15 mg lysate,[64]) macrodomains for 2 h at 4 °C. The enriched samples were then prepared for MS analysis as described previously[32].

## Liquid chromatography and mass spectrometry analysis

Whole proteome LC-MS/MS analyses were performed either on Orbitrap Fusion Lumos (mouse skeletal muscle) or an Orbitrap Q Exactive HF (siMacroD1/siMock U2OS) mass spectrometer (Thermo Fisher Scientific), coupled to ACQUITY UPLC liquid chromatographs (Waters). Peptides were loaded onto a commercial MZ Symmetry C18 Trap Column (100 Å, 5 μm, 180 μm × 20 mm, Waters) followed by nanoEase MZ C18 HSS T3 Column (100 Å, 1.8 μm, 75 μm × 250 mm, Waters). Peptides were eluted over 110 or 115 mins at a 300 nL/min flow rate. An elution gradient protocol was used from 2% to 25% B, followed by two steps at 35% B for 5 min and at 95% B for 5 min, respectively. The mass spectrometers were operated in data-dependent mode (DDA), acquiring full-scan MS spectra (300 – 2000 m/z, depending on the instrument used) at a resolution of 120,000 at 200 m/z. Data-dependent MS/MS were recorded in the Orbitrap and HCD fragmentation with 30% fragmentation energy. Only precursors with intensities above 5000 were selected for MS/MS, and the maximum cycle time was set to 3 sec. Charge state screening was enabled. Single, unassigned, and charge states higher than seven were rejected. Precursor

masses previously selected for MS/MS measurement were excluded from further selection for 25 sec.

Identification of ADP-ribosylated peptides from mouse skeletal muscle was performed on an Orbitrap Fusion Lumos mass spectrometer (Thermo Fisher Scientific) coupled to ACQUITY UPLC liquid chromatograph (Waters). The ADP-ribose product-dependent method called HCD-PP-EThcD[65] was applied, which includes high-energy data-dependent HCD, followed by high-quality HCD and EThcD MS/MS when two or more ADP-ribose fragment peaks (136.0623, 250.0940, 348.07091, and 428.0372) were observed in the initial HCD scan. A detailed description of the MS parameters can be found in ref. [65]. Solvent compositions in channels A and B, which were 0.1% formic acid in water and 0.1% formic acid in acetonitrile, respectively. Peptides were loaded onto loaded on a commercial MZ Symmetry C18 Trap Column (100 Å, 5 µm, 180 µm × 20 mm, Waters) followed by nanoEase MZ C18 HSS T3 Column (100 Å, 1.8 µm, 75 µm × 250 mm, Waters). Peptides were eluted over 110 min at a 300 nL/min flow rate. An elution gradient protocol was used from 2% to 25% B, followed by two steps at 35% B for 5 min and at 95% B for 5 min, respectively.

All relevant data have been deposited to the ProteomeXchange Consortium via the PRIDE partner repository with the data set identifier PXD020281.

## ADP-ribosylome Data Analysis

MS and MS/MS spectra were converted to Mascot generic format (MGF) using Proteome Discoverer, v2.1 (Thermo Fisher Scientific, Bremen, Germany). For the multiple fragmentation techniques (HCD and EThcD) utilized, separate MGF files were created from the raw file for each type of fragmentation. Mascot searches were carried out as previously described[65] with the following protocol modifications. The MGFs were searched against the UniProtKB mouse database (taxonomy 10090, version 20160902), which included 24,905 Swiss-Prot, 34,616 TrEMBL entries, 59,783 decoy hits, and 262 common contaminants. Cysteine carbamidomethylation was set as a fixed modification, and protein N-terminal acetylation and methionine oxidation were set as variable modifications. Finally, S, R, K, D, E, H, C, T and Y residues were set as variable ADP-ribose acceptor amino acids. The neutral losses of 249.0862 Da, 347.0631 Da, and 583.0829 Da from the ADP-ribose were scored in HCD fragment ion spectra[66].

## Protein identification and label-free protein quantification

The acquired whole proteome raw MS data were processed using MaxQuant (version 1.6.2.3), followed by protein identification using the integrated Andromeda search engine[67]. Spectra were searched against a Swissprot Mus musculus reference proteome (taxonomy 10090, version from 2019-07-09) or the Swissprot human reference proteome (taxonomy 9609, version from 2019-07-09), concatenated to its reversed decoyed fasta database and common protein contaminants. Carbamidomethylation of cysteine was set as a fixed modification, while methionine oxidation and N-terminal protein acetylation were set as variables. Enzyme specificity was set to trypsin/P, allowing a minimal peptide length of 7 amino acids and a maximum of two missed cleavages. MaxQuant Orbitrap default search settings were used. The maximum false discovery rate (FDR) was set to 0.01 for peptides and 0.05 for proteins. Label-free quantification was enabled, and a 2 min window for a match between runs was applied. In the MaxQuant experimental design template, each file is kept separate in the experimental design to obtain individual quantitative values. Protein fold changes were computed based on Intensity values reported in the proteinGroups.txt file. A set of functions implemented in the R package SRMService[68] was used to filter for proteins with 2 or more peptides, allowing for a maximum of 4 missing values, and to normalize the data with a modified robust z-score transformation and to compute p-values using the t-test with pooled variance. If all protein measurements are missing in one of the conditions, a pseudo-fold change was computed, replacing the missing group average with the mean of 10% smallest protein intensities in that condition.

## Extraction of intracellular metabolites

Intracellular metabolites were extracted as described[69]. Briefly, the medium was removed, and cells were washed with 0.9% NaCl. Afterward, 200 µL ice-cold methanol and ddH$_2$O containing 2 µg/mL D6-pentanedioic acid as internal standard (IS) were added to the well, and cells were scraped on a cold plate. The liquid was transferred to a 2 mL reaction tube containing 200 µL chloroform. The mixture was vortexed for 10 min at 4 °C and 2000 rpm and subsequently centrifuged at 4 °C and 17,000 × $g$ for 5 min. 200 µL of the upper, polar phase were transferred to a glass vial and dried overnight in a refrigerated vacuum centrifuge at 4 °C.

## Derivatization and GC-MS measurement for metabolome analysis

Before GC-MS measurement, the dried extracts were derivatized using a Gerstel MPS. Dried polar metabolites were dissolved in 15 µL of 2% methoxyamine hydrochloride in pyridine and incubated for 90 min at 40 °C under continuous shaking. After that, 15 µL of N-methyl-N-trimethylsilyl-fluoroacetamide was added and incubated for 30 min at 40 °C under constant shaking. 1 µL of the sample was injected into an SSL injector at 270 °C in splitless mode. Gas chromatography was performed using an Agilent 7890 A GC with a 30 m DB-35MS capillary column (0.25 mm internal diameter, 0.25 µm film thickness). We used helium as the carrier gas at a 1.0 mL/min flow rate. The GC-MS oven temperature was held at 80 °C. After 6 min at 80 °C, temperature increased to 300 °C at a 6 °C/min rate. When 300 °C was reached, the temperature was further increased to 325 °C at 10 °C/min and held for 4 min. This temperature profile results in a total run time of 59.167 min. The GC was connected to an Agilent 5975 C inert XL MSD. The transfer line temperature was set to 280 °C, and the MSD operated under electron ionization at 70 eV. The MS source was held at 230 °C and the quadrupole at 150 °C. Full scan mass spectra were acquired from m/z 70 to m/z 700 at a scan rate of 5.2 scans/sec and a solvent delay of 5 min. GC-MS chromatograms were analyzed using the MetaboliteDetector software package[70].

## Derivatization and GC-MS measurement for metabolic flux analysis

Derivatization and GC-MS measurements were performed as described earlier[71]. GC-MS chromatograms were analyzed using the MetaboliteDetector software package[70], and chemical formulas for calculating mass isotopomer distributions (MIDs) were taken from[72].

## Targeted metabolomics

Cells were plated at 100,000 cells/well in six-well plates in full media and reverse siRNA transfections were performed as described above. Per siRNA (siMock and siMacroD1) 4 wells were prepared and transfected individually. After 48 h, the cells were gently washed with room temperature phosphate-buffered saline (PBS), transferred to ice and 1.5 mL of ice-cold 80:20 MeOH:H$_2$O solution was added to each well. The cells were scraped and transferred to a pre-cooled Eppendorf tube, snap-frozen in liquid nitrogen and thawed in ice before being centrifuged at 16,000 × $g$ for 10 min at 4 °C. Cell extracts were dried downs using a nitrogen evaporator. The dry residue was reconstituted in 50 µL of water. 10 µL of sample extract was mixed with 10 µL of isotopically labeled internal standard mixture in high-performance liquid chromatography vial and used for LC-MS/MS analysis. A 1290 Infinity II UHPLC system (Agilent Technologies) coupled with a 6470 triple quadrupole mass spectrometer (Agilent Technologies) was used for the LC-MS/MS analysis. The chromatographic separation for samples was carried out on a ZORBAX RRHD Extend-C18, 2.1 × 150 mm,

1.8 μm analytical column (Agilent Technologies). The column was maintained at a temperature of 40 °C and 4 μL of sample was injected per run. The mobile phase A was 3% methanol (v/v), 10 mM tributylamine, 15 mM acetic acid in water, and mobile phase B was 10 mM tributylamine, 15 mM acetic acid in methanol. The gradient elution with a flow rate 0.25 mL/min was performed for a total time of 24 min. Afterward, a back flushing of the column using a 6port/2-position divert valve was carried out for 8 min using acetonitrile, followed by 8 min of column equilibration with 100% mobile phase A. The triple quadrupole mass spectrometer was operated in an electrospray ionization negative mode, spray voltage 2 kV, gas temperature 150 °C, gas flow 1.3 L/min, nebulizer 45 psi, sheath gas temperature 325 °C, and sheath gas flow 12 L/min. The metabolites of interest were detected using a dynamic multiple reaction monitoring (MRM) mode. The MassHunter 10.0 software (Agilent Technologies) was used for the data processing. Ten-point linear calibration curves with internal standardization were constructed for the quantification of metabolites.

### Reporting summary

Further information on research design is available in the Nature Portfolio Reporting Summary linked to this article.

## Data availability

The data that support the findings of this study are available in the main text, the supplementary materials and from the corresponding author upon reasonable request. Source data are provided with this paper. The mass spectrometry data used in this study have been deposited to the ProteomeXchange Consortium via the PRIDE partner repository with the data set identifier PXD020281. Source data are provided with this paper.

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

## Acknowledgements

We thank Tobias Suter and Anna Keodora (University of Zurich) for the helpful discussions and for providing editorial assistance as well as for technical assistance, respectively. We thank the Center for Microscopy and Image Analysis, the Functional Genomics Center Zurich and the Centre for Flow Cytometry of the University of Zurich, as well as the

molecular discovery platform of CeMM for their services and assistance. We are also grateful to Dr. Giovanni Pellegrini, previously LAMP, for undertaking the mouse phenotyping and to the technical staff of the Histology Laboratory and the Electron Microscopy Unit, Institute for Veterinary Pathology, Vetsuisse Faculty, University of Zurich, for excellent technical support.

ADP-ribosylation research in the laboratory of M.O.H. is funded by the Kanton of Zurich and the Swiss National Science Foundation (grants 31003A_176177, 310030_205202 and IZLIZ3_200237).

## Author contributions

Project conceptualization and administration: M.O.H., A-K.H. (lead). Data curation and Formal analysis: A-K.H. (lead); L.S. (lead metabolomics); D.M.L.P. (lead proteomics). Investigation (specific experiments): A-K.H. (lead), L.P.F., A.R.S., L.M., E.F. and F.V. (supporting); L.S. (lead metabolomics); D.M.L.P. (lead proteomics); U.H (lead, TEM); F.P. (muscle pathology and morphometry). Visualization and validation: A-K.H. (lead); D.M.L.P. and L.P.F. (supporting). Methodology: A-K.H. (lead), L.P.F. and A.R.S. (supporting); L.S. (lead metabolomics), and A-K. H. (supporting); D.M.L.P (lead proteomics) and A-K.H. (supporting), U.H. (lead TEM); F.P. (lead muscle pathology) and A-K.H. and A.K. (supporting). Writing, review & editing of MS: A-K.H., M.O.H (lead), D.M.L.P, L.P.F., F.P. and A.K. (supporting); A.R.S., L.S., U.H., F.V. K.H. and C.S. (editing).

## Competing interests

The authors declare no competing interest.
