## [Transparent Peer Review file · Nature Communications]

MacroD1 sustains mitochondrial integrity and oxidative metabolism

Corresponding Author: Professor Michael Hottiger

Version 0:

Reviewer comments:

Reviewer #1

(Remarks to the Author)

In the study, Hopp AK et al. demonstrated that loss of MacroD1 decreases oxidative metabolism in mouse muscle. Mechanistically, they found that MacroD1 knockdown leads to mtROS mediated cell death and mitochondrial fragmentation that impairs mitochondrial integrity. Furthermore, metabolomics and proteomics reveal that loss of MacroD1 redirects the carbohydrate flux from the TCA cycle to the pentose-phosphate-pathway.

Major comments,

1. Žaja et al. have demonstrated that MacroD1 is most prevalent in mitochondria of skeletal muscle, and loss of MacroD1 leads to disruption of mitochondria morphology. Therefore, the article is the lack of novelty.
2. The evidence that drew conclusion is not sufficient. For instance, the authors claim that stable isotope assisted metabolomic profiling reveals that loss of MacroD1 rewires the metabolite flux from glucose to the pentose-phosphate cycle rather than into the tricarboxylic acid cycle in summary section, however, there is no data of isotope labelled PPP intermediates in results section. Moreover, based on only NAD⁺ levels to reach the conclusion that MacroD1 regulates cell NADH/ NAD⁺ ratio, it is not rigorous.
3. The conclusion should be drawn through multiple experimental designs. It is very hasty that the conclusion was got through a single experimental method.
4. The content of Material and Methods is so simple, it is very difficult for readers to repeat the results.

Specific points:

1. The evidence that MacroD1 is important for oxidative metabolism is not enough. Some experiments need to be added, such as “quantification of mtDNA Copy Number” and “OCR Using a Seahorse Flux Analyzer”.
2. Authors demonstrated that loss of MacroD1 reduced the number of mitochondria. The authors need to describe the process of obtaining results.
3. Only Fig. 3a is too thin to confirm mitochondria fragmentation. Mitochondria length and mitochondria fission proteins should be examined.
4. Authors demonstrated that knockdown of MacroD1 induced apoptosis. Apoptotic proteins should be examined to further strengthen the reliability of the conclusion.
5. In Fig. 2b, 3g, except quantitative results, representative images need to be shown.
6. In Fig. 3c, MitoSOX fluorescence should be displayed so that readers can visually observe the results.
7. In suppl. Fig 5a, the heat map should show all different metabolites, and the focused metabolites can be marked with *.
8. In Fig. 4d, why was the labeling efficiency of TCA intermediate metabolites so low in cells labeled with ¹³C-glucose as a substrate? except citrate, other metabolites are only significantly different in M2, and there is no difference in M4. What are the possible reasons?
9. Line 8, “8057 Zurich; Switzerland” should be added.
10. Some writings are not consistent (such as “wildtype” and “wild-type”).
11. In Fig. 2d, the magnification should be noted.
12. Line 198, correlate instead of “correlated”.
13. Line 254, “cells” should be added behind “C2C12”.

Reviewer #2

(Remarks to the Author)

The present work analyzed the role of the MacroD1 in mitochondrial function. By using loss of function experiments the authors showed that inhibition of MacroD1 resulted in mitochondrial fragmentation, increased oxidative stress and apoptotic cell death. To get insights they check the ADP-ribosylated proteins by mass-spectrometry in MacroD1 deficient muscles and found a slight reduction. By cross referencing the data with the published MacroD1 interactome and by using bioinformatic tools the authors found a connection with TCA cycle and OXPHOS. Metabolomic analyses confirmed the TCA cycle was altered in MacroD1 knocked down cells. Proteomic analyses showed a reduction of proteins belonging to pyruvate dehydrogenase when MacroD1 was inhibited. Finally, these alterations result in an increased anaerobic glycolysis and improvement insulin sensitivity and glucose tolerance. The data are interesting, and the findings are novel. However, there are some issues that must be addressed to sustain author's claim. The authors should consider the following points.

Point 1. Fig1C: is the difference significant? Otherwise it is not justified to claim that the limb muscle weight is reduced in KO mice. These mice should be better characterized in terms of fiber type distribution for myosins (slow versus fast) and mitochondria (beta-oxidative fibers versus glycolytic) as well as fiber size distribution. Authors should also look at the single muscles such as gastrocnemius, tibialis anterior, soleus and EDL.

Point 2. Suppl Fig2C. The magnification of the pictures is too low and it is difficult to see the staining. Show picture with the same magnification of Suppl Fig 2B.

Point3. Fig2b. The decrease of mitochondrial numbers in the EM pictures does not address whether mitochondrial number or mitochondrial mass is reduced. Please quantify the mitochondrialDNA/nuclear DNA (as readout of number) and mitochondrial proteins (e.g. VDAC or TOM20, as readout of mass). It is difficult to claim that the white arrows point on mitochondrial with abnormal cristae. By this magnification they look more similar to vesicles than mitochondria. Please show better and high quality pictures

Point 4. Fig3a. To better understand whether mitochondria are more fragmented or less fused the authors should monitor the mitochondrial shaping proteins (Mifusin1 and 2, OPA1, DRP1 and Fis1, Mff and MiD49 MiD51).

Point 5. Are the apoptotic nuclei and the nuclei with smaller size present in MacroD1-deficient muscles?

Point6. The decrease in TMRE does not mean that mitochondria are depolarized but that they are inefficient in driving protons in the intermembrane space (less polarized).

Point7. Fig 3c. Please check the level of protein oxidation (protein carbonylation). To prove that mitochondria are the sources of ROS that affect mitochondrial fragmentation and cellular viability authors must use mitochondrial-targeted ROS scavenger such as MitoTEMPO. Which is the link between ROS production and mitochondrial fragmentation? Is it mitophagy, which is activated by oxidative stress, involved?

Point8. Fig3g. Is the decrease in nuclear size due to an increase in chromatin condensation due to epigenetic changes?

Point9. Fig.3h is the increase statistically significant? Otherwise it is not justified to claim "... a reproducibly specific increase..."

Point10. The mass spectrometry for ADP-ribosylation should be repeated acutely MacroD1 knocked down cells to avoid the effects of compensation.

Point11. The increase in PPP should generate more NADPH and therefore, more protected by oxidative stress. Are the levels of Glutathions and NADPH affected?

Point12. Fig 4b. What does it mean the M0-M1-M2-M3 of X-axis? I have not found any explanation either in figure legends or text

Point13. Fig4e. Lane 290: Glutamine deprivation does not increase cell death in siMacroD1 cells. Please correct the text

Point14. Which is the connection between MacroD1 oxidative stress and AKT? Please clarify

Point15. The proteomic analyses on MacroD1 KO muscle and siMacroD1 cells showed that mitochondrial proteins are down. However, the network is fragmented and therefore it is expected that mitochondrial mass is reduced. It would be important to have data on mitochondrial number and mass (see above) and normalize the different downregulated proteins for mitochondrial mass.

Point16. Fig6. Are the effects on glucose and insulin tolerance test gender specific? Please check also males. Authors should also test gluconeogenesis by using the pyruvate test

Version 1:

Reviewer comments:

Reviewer #1

(Remarks to the Author)

The study by Ann-Katrin Hopp et al., entitled "MacroD1 sustains mitochondrial integrity and oxidative metabolism" describes the role of MacroD1, a mono-ADP-ribosylhydrolase localized to mitochondria, in regulating metabolic homeostasis. The authors have provided a comprehensive and detailed response to the reviewers' comments, significantly enhancing the rigor and completeness of the study through additional experiments, optimized data analysis, and text revisions. The newly added data, including targeted metabolomics, Seahorse analysis, mitochondrial morphological quantification, and PARP1/Caspase 3 detection, effectively support the core conclusions and address most of the raised concerns.

Major comments:

1. Although the manuscript demonstrates that MacroD1 deficiency leads to mitochondrial damage and metabolic reprogramming, the causal relationship between these two phenomena is not fully explored. Can intervening in metabolic

pathways mitigate mitochondrial damage?

2. MacroD1 may regulate mitochondrial function by affecting ADP-ribosylation, but the specific molecular mechanisms remain unclear. Does the enzymatic activity of MacroD1 affect the activity of key metabolic enzymes through specific substrates (e.g., ADP-ribosylated proteins)?

3. The newly added NADPH/NADP+ data only show static ratios and do not reveal their dynamic responses under metabolic stress.

4. The authors have added experiments on mitochondrial DNA copy numbers, the assessment of mitochondrial quality seems still incomplete. Can further analysis be conducted on mitochondrial membrane potential and the activity of mitochondrial respiratory chain complexes to more comprehensively evaluate changes in mitochondrial function?

5. The authors mentioned the potential role of MacroD1 in cancer but did not delve into its association with the Warburg effect or the therapeutic prospects of combining metabolic inhibitors.

Minor comments:

1. The resolution of the EM images in Suppl. Fig. 2C is low. It is recommended to provide higher-magnification zoomed-in images to clearly show the abnormal mitochondrial cristae structures.

2. Some experiments (e.g., IPGTT/IPITT) do not specify whether multiple comparisons were corrected. The statistical methods need to be clarified.

Reviewer #2

(Remarks to the Author)

The authors addressed most of my concerns. The new version is improved

Version 2:

Reviewer comments:

Reviewer #1

(Remarks to the Author)

Authors have addressed all of my issues.

Point-by-point response to reviewer comments

We would like to thank the editor and the reviewer for thoroughly reviewing our manuscript. We have carefully considered all comments as well as suggestions, have performed additional experiments and revised the manuscript to address all concerns (see point-by-point response below). We believe the revised manuscript is much improved. We hope you will find the revised manuscript acceptable for publication in *Nature Communication*.

Summary of figure changes:

Figure(s) in the revised manuscript	Revised	New	unchanged	Corresponding figure(s) in original manuscript
F1A			X	F1A
F1B			X	F1B
F1C			X	F1D
F1D			X	F1E
F1E			X	F1F
F1F			X	F1G
F2A			X	F2A
F2B		X		
F2C		X		
F2D		X		
F2E			X	F2B
F2F	X			F2C
F2G	X			F2D
F3A	X			F3A
F3B		X		
F3C			X	F3B
F3D		X		
F3E	X			F3C
F3F			X	F3D
F3G	X			F3E
F3H		X		
F3I			X	F3F
F3J	X			F3G
F4A			X	F4A
F4B			X	F4B
F4C	X			F4D
F4D		X		
F4E	X			F4E
F5A		X		
F5B		X		
F5C		X		
F5D		X		
F5E		X		
F5F		X		
F5G			X	F4C

F5H			X	F4C
F6A			X	F6A
F6B			X	F6B
F6C			X	F6C
F6D			X	F6D
SF1A			X	SF1A
SF1B			X	SF1B
SF1C			X	SF1C
SF1D			X	SF1D
SF1E			X	SF1E
SF1F			X	SF1F
SF1G			X	SF1G
SF1H			X	SF1H
SF1I			X	SF1I
SF1J		X		
SF2A		X		
SF2B		X		
SF2C			X	SF2D
SF2D		X		
SF3A			X	
SF3B			X	F3A
SF3C		X		
SF3D	X			SF3B
SF3E	X			SF3C
SF3F	X			SF3D
SF3G			X	SF3E
SF3H			X	SF3F
SF3I		X		
SF3J		X		
SF3K			X	SF3G
SF4A			X	SF4A
SF4B	X			SF4B
SF4C			X	SF4C
SF4D			X	SF4D
SF5A			X	SF5G
SF5B			X	F5A
SF5C			X	F5C
SF5D		X		
SF5E			X	F3E
SF5F		X		
SF6A			X	F5D
SF6B			X	F5B
SF7A			X	SF5A
SF7B			X	SF5B
SF7C			X	SF5C
SF7D			X	SF5D
SF7E			X	SF5E
SF7F	X			SF5F
SF8A			X	F3H
SF8B			X	SF4E

SF8C			X	SF4F
SF8D			X	F3I
SF8E	X			SF4G
SF8F		X		

Reviewer #1 (Remarks to the Author):

In the study, Hopp AK et al. demonstrated that loss of MacroD1 decreases oxidative metabolism in mouse muscle. Mechanistically, they found that MacroD1 knockdown leads to mtROS-mediated cell death and mitochondrial fragmentation that impairs mitochondrial integrity. Furthermore, metabolomics and proteomics reveal that loss of MacroD1 redirects the carbohydrate flux from the TCA cycle to the pentose-phosphate-pathway.

Major comments:

1. Žaja et al. have demonstrated that MacroD1 is most prevalent in mitochondria of skeletal muscle, and loss of MacroD1 leads to disruption of mitochondria morphology. Therefore, the article is the lack of novelty.

Response: We would kindly like to clarify that the focus of this manuscript was not to demonstrate that MacroD1 is predominantly expressed in skeletal muscle. This is indeed a published fact upon which we based our experiments. This is also the reason why we mentioned the publication by Žaja et al. in our introduction.

The goal of our work was to gain further functional insight into the role of MacroD1 in mitochondria, particularly in skeletal muscle, building on previous findings mentioned above and in other older papers. While we could demonstrate here that MacroD1 has a relevant role in muscle tissue, it seems to have similar functions independent of cell type and species. We believe that the high expression of MacroD1 in muscle is, in fact, a consequence of the high mitochondrial content of this tissue. Moreover, although Žaja et al. already described mitochondrial fragmentation, we provide further new and important information on the underlying reason. In fact, we could show that MacroD1 is important not only for mitochondrial oxidative metabolism but also for regulating the cell's overall redox state. It is also noteworthy that this phenomenon is not only observed *in vitro* but appears to be equally relevant *in vivo*.

2. The evidence that drew conclusion is not sufficient. For instance, the authors claim that stable isotope assisted metabolomic profiling reveals that loss of MacroD1 rewires the metabolite flux from glucose to the pentose-phosphate cycle rather than into the tricarboxylic acid cycle in summary section, however, there is no data of isotope labelled PPP intermediates in results section. Moreover, based on only NAD⁺ levels to reach the conclusion that MacroD1 regulates cell NADH/ NAD⁺ ratio, it is not rigorous.

Response: As the metabolomics approach taken initially could not capture intermediates of the pentose-phosphate pathway, we have now included a different targeted metabolomics method that measures approximately 140 metabolites, mainly from the central carbon metabolism. This approach revealed a significant increase in the pentose-phosphate pathway intermediates and various nucleotides upon MacroD1 knockdown (new Fig. 5A-D). In addition, we have now included direct NADPH/NADP⁺ measurements to validate and confirm an increase in the NADPH/NADP⁺ ratio upon knockdown of MacroD1 (new Fig. 5E).

Additionally, the newly included metabolomics data also captured a significant increase in glutathione (new Fig. 5D), thus giving a rationale for why MacroD1 knockout would lead to an increase in the pentose-phosphate pathway flux. Together, these data support the cell-based observation that lack of MacroD1 increases ROS and that cells try to mitigate this by supporting the production of antioxidants.

3. The conclusion should be drawn through multiple experimental designs. It is very hasty that the conclusion was got through a single experimental method.

Response: We thank the reviewer for the valuable feedback. Firstly, we would like to highlight, that already in the first version of the manuscript we provided several experimental approaches hinting into the direction of a redirection into the PPP: the untargeted metabolomics approach already revealed a significant increase in several intermediates of the PPP, including fructose, mannose, ribitol, and myo-inositol, but also serine (Suppl. Fig. 7A). Furthermore, the isotope tracing revealed also an increase in serine. The newly included targeted metabolomics approach now also captured a significant increase in intermediates of the pentose-phosphate pathway, as well as various nucleotides (new Fig. 5A-D) upon knockdown of MacroD1. In addition, we also observed an increase in glutathione. This validated our cell-based assays, showing an increase in superoxide production, and allowed us to conclude that the increase in the pentose-phosphate pathway is likely orchestrated to support the cell's antioxidant defense (e.g., to generate NADPH and glutathione). Similarly, the proteomics data that were part of the first version of the manuscript captured an increase in proteins involved in the cellular antioxidant response. These are also in line with our new metabolomics measurements, as well as with our cell-based assays. Thirdly, in line with the increase in the pentose-phosphate pathway, we could now also directly show an increase in the NADPH/NADP⁺ ratio when MacroD1 was knocked down (new Fig. 5E). At the same time, we could not directly capture intermediates of the PPP with our labeling approach, however, this method also captured the increase in serine (Suppl. Fig. 7B), another metabolite indicative of increased pentose-phosphate pathway activity.

In summary, we provide further evidence for the increase in the pentose-phosphate pathway flux and the mechanisms underlying this observation by metabolomics and direct NADPH/NADP⁺ measurements. Thus, we believe that the different experimental designs support our original conclusion very well through various methods and from different angles.

4. The content of Material and Methods is so simple, it is very difficult for readers to repeat the results.

Response: We apologize for the initial brevity of this section. We extended the Material and Methods section with more details to facilitate the reproduction of the provided results by the community (pages 23 ff).

Specific points:

1. The evidence that MacroD1 is important for oxidative metabolism is not enough. Some experiments need to be added, such as “quantification of mtDNA Copy Number” and “OCR Using a Seahorse Flux Analyzer”.

Response: We appreciate the reviewer's valuable comment. To better understand the changes in mitochondrial metabolism, we conducted Seahorse measurements after the knockdown of MacroD1 (new Fig. 4D). As suggested by the proteomics data (e.g., a reduction of mitochondrial proteins involved in oxidative metabolism; new Suppl. Fig. 5B and 6A and B) and supported by the metabolic flux

analysis and the untargeted metabolomics (Fig. 4C and Suppl. Fig. 7A), the knockdown of MacroD1 decreases both basal as well as maximal respiration, demonstrating that lack of MacroD1 reduces OXPHOS in general.

In mice, we have undertaken an additional experiment to determine in detail the changes in myofiber size, mitochondrial content, and mitochondrial structure in the muscles of the MacroD1^{-/-} mice (new Fig. 2B-D and Suppl. Fig. 1J, 2A and 2B). The results confirm and further specify the findings reported in the initial manuscript.

In addition, the decrease in mitochondrial oxidative metabolism is further supported by the qPCR data (Fig. 1F), as well as the running belt experiments (Fig. 1C-E) that have already been present in the first version of the manuscript.

Furthermore, we determined the mitochondrial DNA content in MacroD1^{-/-} and wildtype mice (new Suppl. Fig. 2D) and observed no change. The proteomics data shown in Fig. 6 do however reveal a reduction in specific mitochondrial proteins (primarily those involved in oxidative metabolism), but not all. Together with the ultrastructural findings (Fig. 2F), our data point towards the fact that in mice lacking MacroD1, mitochondrial quality, rather than quantity, is considerably changed. This is further supported by the results of the new analysis of muscle fiber size and distribution (new Fig. 2B and C). As discussed in the manuscript (p. 8), despite the fact that acute removal of MacroD1 is lethal in most cell types (Fig. 3), the animals are fully viable, pointing towards an adaptation that happened during their generation process. We, therefore, don't expect mitochondrial and metabolic changes to be as drastic in the animals as we observe them to be for cells that experience an acute knockdown of MacroD1.

2. Authors demonstrated that loss of MacroD1 reduced the number of mitochondria. The authors need to describe the process of obtaining results.

Response: As per the reviewer's request, we have provided a more detailed description of the use of MitoTrackerTM Green (page 26). Additionally, we have revised the descriptions of the two dyes, MitoSOXTM and TMRE (page 27), and how these different assays allow for a determination of the mitochondrial load, mitochondrial superoxide production, and mitochondrial polarization, respectively.

3. Only Fig. 3a is too thin to confirm mitochondria fragmentation. Mitochondria length and mitochondria fission proteins should be examined.

Response: We appreciate the reviewer's valuable input. In addition to the double-blinded manual assessment of the mitochondria's fission/fusion state (that has now been moved to the supplementary material: Suppl. Fig. 3B and Suppl. Fig. 5E), we have performed unbiased, quantitative image analysis using the FIJI plugin "Mitochondrial Analyzer" (new Fig. 3B and H). With this approach, mitochondria from single cells were first skeletonized (new Suppl. Fig. 3C), and subsequently, parameters such as mitochondrial form, branch length, and branch size were determined in an automated manner. In line with our manual assessment, the knockdown of MacroD1 altered the mitochondrial shape (new Fig. 3B, left panel) and also reduced both the lengths and the number of branches per mitochondrion (new Fig. 3B, mid and right panel). Most importantly, all three parameters re-increased significantly after treatment with the antioxidant NAC (new Fig. 3H). Here, as well, data were in line with our previous double-blinded manual assessment (now Suppl. Fig. 5E).

4. Authors demonstrated that knockdown of MacroD1 induced apoptosis. Apoptotic proteins should be examined to further strengthen the reliability of the conclusion.

Response: As per the reviewer's request, we assessed PARP1 cleavage and Caspase 3 expression after the knockdown of MacroD1. This analysis provided additional evidence that the knockdown of MacroD1 indeed induced an increase in cleaved PARP1 (new Suppl. Fig. 3I) and Caspase 3 (new Suppl. Fig. 3J), supporting that knockdown of MacroD1 indeed induces apoptosis.

5. In Fig. 2b, 3g, except quantitative results, representative images need to be shown.

Response: We apologize for the lack of clarity regarding former Fig. 2B (now Fig. 2E in the revised manuscript). Respective images were already shown in the original version of the manuscript (old Suppl. Fig. 2D, now Suppl. Fig. 2C). Further, representative images have been added to the revised version of the manuscript for former Fig. 3G (now Fig. 3J).

6. In Fig. 3c, MitoSOX fluorescence should be displayed so that readers can visually observe the results.

Response: We would like to clarify that the graphs in question were generated from a flow cytometric experiment, as recommended by the manufacturer. To avoid any confusion, we have updated the corresponding section of the methods (page 27). Furthermore, we have included representative flow cytometry-derived histograms (new Fig. 3E) to aid in visualizing the results and differences. Additionally, we now provide IF pictures in the supplementary material (new Suppl. Fig. 5D).

7. In suppl. Fig 5a, the heat map should show all different metabolites, and the focused metabolites can be marked with *.

Response: We would like to clarify that the heatmap in the former Suppl. Fig. 5A (now Suppl. Fig. 7A) derives from an untargeted metabolomics experiment. In many cases, detected masses cannot unambiguously be assigned to a specific metabolite and metabolite identity remains unknown. To provide a better overview, we kept the row names only when a clear assignment was possible.

8. In Fig. 4d, why was the labeling efficiency of TCA intermediate metabolites so low in cells labeled with ^{13}C -glucose as a substrate? except citrate, other metabolites are only significantly different in M2, and there is no difference in M4. What are the possible reasons?

Response: We thank the reviewer for highlighting this interesting detail. In the experiment, the cells were cultured in 100% $^{13}\text{C}_6$ -Glucose for 36 hours which is commonly assumed to be more than enough time to reach isotopic steady state in TCA cycle metabolites (commonly reached after ~ 2 hours¹). The data indicate that the contribution from glucose to most TCA cycle metabolites is limited. The non-labeled fraction M0 is over all quite high ($>70\%$ for Glu, Succ, Fum, Mal, Asp), which is further increased to $M0 > 80\%$ when MacroD1 is knocked down. However, parallel data with $^{13}\text{C}_5$ -glutamine labeling result in rather small non-labelled fractions ($M0 < 20\%$) indicating that glutamine is the main substrate contributing to the metabolite pool of the mentioned TCA cycle metabolites in C2C12 cells. (revised Fig. 4C).

9. Line 8, "Zurich; Switzerland" should be added.

Response: We would like to thank the reviewer for her/his attention. The missing information has been added to the revised version of the manuscript.

10. Some writings are not consistent (such as “wildtype” and “wild-type”).

Response: We would like to thank the reviewer for their attention. This was adjusted in the revised version of the manuscript.

11. In Fig. 2d, the magnification should be noted.

Response: We apologize for this omission. Information on the magnification (scale bars) is included for all images in the revised manuscript.

12. Line 198, correlate instead of “correlated”.

Response: This has been corrected in the revised version of the manuscript.

13. Line 254, “cells” should be added behind “C2C12”.

Response: This has been corrected in the revised version of the manuscript.

Reviewer #2 (Remarks to the Author):

The present work analyzed the role of the MacroD1 in mitochondrial function. By using loss of function experiments the authors showed that inhibition of MacroD1 resulted in mitochondrial fragmentation, increased oxidative stress and apoptotic cell death. To get insights they check the ADP-ribosylated proteins by mass-spectrometry in MacroD1 deficient muscles and found a slight reduction. By cross referencing the data with the published MacroD1 interactome and by using bioinformatic tools the authors found a connection with TCA cycle and OXPHOS. Metabolomic analyses confirmed the TCA cycle was altered in MacroD1 knocked down cells. Proteomic analyses showed a reduction of proteins belonging to pyruvate dehydrogenase when MacroD1 was inhibited. Finally, these alterations result in an increased anaerobic glycolysis and improvement insulin sensitivity and glucose tolerance. The data are interesting, and the findings are novel. However, there are some issues that must be addressed to sustain author’s claim. The authors should consider the following points.

Point 1. Fig1C: is the difference significant? Otherwise it is not justified to claim that the limb muscle weight is reduced in KO mice. These mice should be better characterized in terms of fiber type distribution for myosins (slow versus fast) and mitochondria (beta-oxidative fibers versus glycolytic) as well as fiber size distribution. Authors should also look at the single muscles such as gastrocnemius, tibialis anterior, soleus and EDL.

Response: We agree with the reviewer that the mice were not fully worked up. In order to address the shortcomings of the *in situ* part of the initial manuscript, we have undertaken an additional standardized investigation on a larger cohort of MacroD1^{-/-} and wildtype animals (each 5 females and 5 males, aged 6-7 weeks). We have investigated the following parameters:

- Morphologic features of the M. quadriceps femoris, M. gastrocnemius, M. triceps brachii and M. soleus, and the myocardium (left ventricular free wall), using cryosections stained with hematoxylin eosin (new Suppl. Fig. 1J).
- Fiber size distribution in all examined skeletal muscles (new Fig. 2B).
- Fiber type distribution (immunohistochemistry for myosin) (new Fig. 2C)
- Succinic acid dehydrogenase (SDH) activity staining as a marker for mitochondria (new Fig. 2D).
- Ultrastructural examination of M. triceps brachii and myocardium (transmission electron microscopy).

These investigations have provided the following results:

- Cryosections stained with H&E revealed no morphological changes in MacroD1^{-/-} mice.
- The muscle fibers were significantly smaller in diameter in M. quadriceps femoris and M. triceps brachii of MacroD1^{-/-} mice than in the controls (p<0.0001 for both muscles). No difference in fiber diameter in M.gastrocnemius and M.soleus was observed (p=0.0566 and p=0.2345, respectively).
- Trend to lower SDH activity in MacroD1^{-/-} mice compared to wildtype but without significant difference, while *Sdha* expression was slightly but significantly reduced (Suppl. Fig. 1I).
- Immunohistochemistry for type I myofiber-specific myosin in M.soleus showed a significantly higher proportion of MyHC7-negative fibers (i.e. type II glycolytic fibers) in MacroD1^{-/-} mice compared to wild type (p<0.0001).
- Deformation and lysis of mitochondrial cristae (TEM).

The manuscript has been updated to incorporate this additional investigation (M&M, results, discussion sections, and new or replaced Figures).

Point 2. Suppl Fig2C. The magnification of the pictures is too low and It is difficult to see the staining. Show picture with the same magnification of Suppl Fig 2B.

Response: Given that the mitochondrial localization of MacroD1 has well been characterized in literature ^{2,3} and that this manuscript really focusses on its functional characterization, we have removed this part.

Point3. Fig2b. The decrease of mitochondrial numbers in the EM pictures does not address whether mitochondrial number or mitochondrial mass is reduced. Please quantify the mitochondrialDNA/nuclear DNA (as readout of number) and mitochondrial proteins (e.g. VDAC or TOM20, as readout of mass). It is difficult to claim that the white arrows point on mitochondrial with abnormal cristae. By this magnification they look more similar to vesicles than mitochondria. Please show better and high quality pictures

Response: As part of the additional investigation of MacroD1^{-/-} and wildtype mice, we have also undertaken a more in-depth ultrastructural examination. The new images better illustrate the mitochondrial changes, and at a higher magnification (new Fig. 2F, structurally abnormal mitochondria are marked with black asterisks). In addition, we have also performed mitochondrial DNA quantifications, as suggested by the reviewer. Interestingly, while we see changes in mitochondrial quality and mass (e.g. reduction in mitochondrial proteins as assessed by mass spectrometry, Fig. 6B), we do not observe changes in mitochondrial copy numbers (new Suppl. Fig. 2D), indicating that MacroD1 regulates mitochondrial metabolism and thus quality, rather than mitochondrial DNA quantity and providing important additional information about the function of the hydrolase. Respective results have been discussed in the manuscript accordingly (p. 5).

Point 4. Fig3a. To better understand whether mitochondria are more fragmented or less fused the authors should monitor the mitochondrial shaping proteins (Mtfusin1 and 2, OPA1, DRP1 and Fis1, Mff and MiD49 MiD51).

Response: Based on the mass spectrometry data, the only very significant change we could observe, was an increase in the mitochondrial fission process 1 (MTFP1; Suppl. Fig. 5C) following knockdown of MacroD1. We have performed additional Western blot and qPCR experiments to investigate potential

changes in other mitochondrial fission and fusion proteins (data not shown), but, in line with the mass spec data, couldn't identify further differences.

Point 5. Are the apoptotic nuclei and the nuclei with smaller size present in MacroD1-deficient muscles?
 Response: The histological examination did not reveal any evidence of pathological changes in the muscles of MacroD1^{-/-} mice; in particular, there was no morphological evidence of apoptosis. However, given that the mouse is viable, while transient knockdown of MacroD1 is lethal to most cultured cells, we suggest, as discussed, that the mice have adapted to the lack of MacroD1 during the selection procedure (page 6). In addition, there is evidence that specifically skeletal muscle is precluded from apoptosis to some extent⁴.

Point6. The decrease in TMRE does not mean that mitochondria are depolarized but that they are inefficient in driving protons in the intermembrane space (less polarized).

Response: We thank the Reviewer for pointing this out and we have revised our statements accordingly (page 6).

Point7. Fig 3c. Please check the level of protein oxidation (protein carbonylation). To prove that mitochondrial are the sources of ROS that affect mitochondrial fragmentation and cellular viability authors must use mitochondrial-targetted ROS scavenger such as MitoTEMPO. Which is the link between ROS production and mitochondrial fragmentation? Is it mitophagy, which is activated by oxidative stress, involved?

Response: As requested by the reviewer, we attempted to rescue the observed lethality upon knockdown of MacroD1 using the mitochondrial ROS scavenger MitoTEMPO. Unfortunately, in our hands, prolonged treatment of C2C12 cells with MitoTEMPO was lethal to control cells (see Fig. 1), making it challenging to draw conclusions regarding a potential rescue. MitoSOX, the ROS probe used in this manuscript, is, however, considered very specific for mitochondrial-derived superoxide. This strengthens our conclusion that the observed increase in superoxide is of mitochondrial origin.

Fig. 1: MitoTempo results in increased cell death in C2C12 myoblasts. C2C12 cells were either treated with MitoTempo for 24 to 36h or left untreated, and cell viability was analyzed using annexinV staining and subsequent flow cytometry.

We also thank the reviewers for the idea of assessing the involvement of mitophagy in the observed loss of MacroD1 phenotypes. We performed a double immunofluorescence staining experiment for ATP5A, as a mitochondrial marker, and for LC3 in C2C12 cells and observed both an increase in LC3 signal and a colocalization of LC3 with ATP5A after KD of *MacroD1* (see Fig. 2, next page).

Fig. 2: C2C12 cells were knocked down for MacroD1 and then fluorescently stained for ATP5A and LC3A/B.

This data indeed suggests that lack of MACROD1 leads to enhanced association of mitochondrial material with autophagosomes and thus mitophagy. This is in line with the reduction in overall mitochondrial load, observed following constitutive knockdown of MacroD1 in C2C12 cells (new Suppl. Fig. 4C) and does indeed point towards an increase in mitophagy. We refrained from adding this data to the manuscript as a firm statement would require additional specialized assays (including ultrastructural analysis) which would be beyond the scope of this manuscript.

Point8. Fig3g. Is the decrease in nuclear size due to an increase in chromatin condensation due to epigenetic changes?

Response: It is very interesting that the reviewers have brought up this point. Based on our proteomics data set, we observe indeed many enzymes involved in chromatin architecture to be significantly deregulated (Suppl. Fig. 5B, 5C and 6A and Suppl. Table 1). However, further analysis of these aspects requires substantial investigation and is out of the scope of this manuscript. However, we have included this point in the revised discussion (page 14).

Point9. Fig.3h is the increase statistical significant? Otherwise it is not justified to claim "...a reproducibly specific increase..."

Response: We thank the authors for pointing out the missing significance indication. The knockdown effect in old Fig. 3H (now Suppl. Fig. 8A) is indeed statistically significant.

Point10. The mass spectroscopy for ADP-ribosylation should be repeated acutely MacroD1 knocked down cells to avoid the effects of compensation.

Response: We would like to kindly point out that the knockdown was indeed performed after acute knockdown of MacroD1 (60h post-siRNA transfection). The reason was, indeed, to avoid adaptation and to have a picture of what was happening at the point where all other cell culture experiments were performed. This point has been included in the revised manuscript (page 7).

Point11. The increase in PPP should generate more NADPH and, therefore, more protected by oxidative stress. Are the levels of Glutathions and NADPH affected?

Response: We thank the reviewers for this important aspect. Indeed, according to our newly added data, both glutathione and NADPH levels are affected by the lack/reduction of MacroD1 (new Fig. 5D and

E). In line with a general increase in pentose-phosphate pathway metabolites, we observe a significant increase in glutathione and its precursor glycine (new Fig. 5D) and a concomitant increase in the ratio of NADPH/NADP⁺ (new Fig. 5E).

Point12. Fig 4b. What does it means the M0-M1-M2-M3 of X-axis? I have not found any explanation either in figure legends or text.

Response: We apologize for this lack of information. Briefly, M is short for “mass” and hence refers to the mass of a given metabolite. Upon incorporation of the isotope-labeled carbon atoms, the mass of the metabolite increases according to the number of carbons incorporated. M0 would thus represent a completely unlabeled metabolite, while M1 would represent the metabolite containing 1 heavy-labeled carbon and so on. As described in new Fig. 4A, not every possible mass can be obtained for every metabolite due to how glucose is further metabolized. As an example, two carbons of pyruvate are spiked into the TCA cycle and added to oxaloacetate to generate citrate. Hence, citrate can exist either completely unlabeled (M0), containing 2 labeled C-atoms (M2), 4 (M4), or 6 (M6), but not with only 1 labeled C-atom (M1). We included a better explanation in the figure legend and the results part.

Point13. Fig4e. Lane 290: Glutamine deprivation does not increase cell death in siMacroD1 cells. Please correct the text.

Response: We would like to thank the reviewer for noticing the error. This has been adjusted in the revised version of the manuscript (page 9).

Point14. Which is the connection between MacroD1 oxidative stress and AKT? Please clarify.

Response: The kinase Akt is a known positive regulator of the cellular antioxidant response⁵. In fact, supra-physiological ROS levels were reported to result in an increase in Akt activation that is mediated via Akt phosphorylation, ultimately resulting in the generation of antioxidants. We have probed for phosphorylated AKT as a different readout supporting the hypothesis that lack of MacroD1 increases cellular ROS levels.

Point15. The proteomic analyses on MacroD1 KO muscle and siMacroD1 cells showed that mitochondrial proteins are down. However, the network is fragmented, and therefore, it is expected that mitochondrial mass is reduced. It would be important to have data on mitochondrial number and mass (see above) and normalize the different downregulated proteins for mito mass.

Response: As requested by the reviewer, we have analyzed mitochondrial copy numbers (new Suppl. Fig. 2D) and observed no change between MacroD1^{-/-} and wildtype animals. The proteomics data shown in Fig. 6 do indeed reveal a reduction in specific mitochondrial proteins (primarily those involved in oxidative metabolism), but not all. Together with the ultrastructural findings (Fig. 2F), our data point towards the fact that in mice lacking MacroD1, mitochondrial quality, rather than quantity, is considerably changed. This is further supported by the results of the new analysis of muscle fiber size and distribution (new Fig. 2B and C). As discussed in the manuscript (p. 6), even though acute removal of MacroD1 is lethal in most cell types (Fig. 3), the animals are fully viable, pointing towards an adaptation that happened during their generation process. We, therefore, don't expect mitochondrial and metabolic changes to be as drastic in the animals as we observe them to be for cells that experience an acute knockdown of MacroD1.

Point16. Fig6. Are the effect on glucose and insulin tolerance test gender specific? Please check also males. Authors should also test gluconeogenesis by using the pyruvate test.

Response: We observed no gender-specific difference, and therefore, the graph contains combined data from 4 male and 4 female mice. Since we have no indications that gluconeogenesis is vital for the effect that MacroD1 seems to have on cell and organ physiology, we believe that those animal experiments are out of scope for the central message of this manuscript.

References:

- 1 Jang, C., Chen, L. & Rabinowitz, J. D. Metabolomics and Isotope Tracing. *Cell* **173**, 822-837, doi:10.1016/j.cell.2018.03.055 (2018).
- 2 Agnew, T. *et al.* MacroD1 Is a Promiscuous ADP-Ribosyl Hydrolase Localized to Mitochondria. *Front Microbiol* **9**, 20, doi:10.3389/fmicb.2018.00020 (2018).
- 3 Zaja, R. *et al.* Comparative analysis of MACROD1, MACROD2 and TARG1 expression, localisation and interactome. *Sci Rep* **10**, 8286, doi:10.1038/s41598-020-64623-y (2020).
- 4 Schwartz, L. M. Skeletal Muscles Do Not Undergo Apoptosis During Either Atrophy or Programmed Cell Death-Revisiting the Myonuclear Domain Hypothesis. *Front Physiol* **9**, 1887, doi:10.3389/fphys.2018.01887 (2018).
- 5 Xie, X., Shu, R., Yu, C., Fu, Z. & Li, Z. Mammalian AKT, the Emerging Roles on Mitochondrial Function in Diseases. *Aging Dis* **13**, 157-174, doi:10.14336/AD.2021.0729 (2022).

Response to the feedback of Reviewer 1

Major comments:

1. Although the manuscript demonstrates that MacroD1 deficiency leads to mitochondrial damage and metabolic reprogramming, the causal relationship between these two phenomena is not fully explored. Can intervening in metabolic pathways mitigate mitochondrial damage?

Response: We thank the reviewers for raising this important aspect. Interestingly, while antioxidant treatment could rescue cell viability and mitochondrial fragmentation (Fig. 3 F-H and new Suppl. Fig. 7C), NAC treatment did not revert the decreased incorporation of glucose derivatives into the TCA cycle (new Suppl. Fig. 7D). Similarly, NAC treatment did not rescue the increase in mitochondrial ADP-ribosylation observed after knockdown of MacroD1 (see Fig.1 in this response, not part of the manuscript and for the reviewer only). Together, the data suggest that lack of MacroD1 primarily results in ADP-ribosylation-dependent metabolic reprogramming, which, in unadapted cells, leads to an increased production of mitochondrial ROS, subsequently resulting in mitochondrial damage and cell death (revised text page 8). Further, control C2C12 cells show a slight yet significant susceptibility towards glutamine starvation that is not further enhanced after the knockdown of MacroD1 (Fig. 4E). As, in most cell culture cells, glutamine directly feeds into the TCA cycle to support OXPHOS (extensively reviewed in Yoo *et al.*, *Exp Mol Med* **52**:1496 (2020)), this observation is in line with our hypothesis that MacroD1 supports mitochondrial respiration because MacroD1 deficient cells that already show severe OXPHOS impairments (Fig. 4C/D) do not further care about a lack of glutamine.

Fig. 1: Mitochondrial ADP-ribosylation after MacroD1 knockdown. MacroD1 was knocked down in C2C12 cells, treated or not with NAC, and cells were subsequently fixed at 48h post-siRNA transfection to analyze mitochondrial ADP-ribosylation via IF.

2. MacroD1 may regulate mitochondrial function by affecting ADP-ribosylation, but the specific molecular mechanisms remain unclear. Does the enzymatic activity of MacroD1 affect the activity of key metabolic enzymes through specific substrates (e.g., ADP-ribosylated proteins)?

Response: We thank the reviewer for raising this interesting aspect, that we have indeed been thinking about a lot. First, we would like to refer to the evidence provided in the manuscript showing that the enzymatic activity of MacroD1 is important for the mitochondrial and metabolic phenotypes we see in cell cultures, as only complementation with wildtype, but not mutant MacroD1 results in a rescue (Fig. 3I/J and Suppl. Fig. 5F). However, the ADP-ribosylome analysis we performed on muscle tissue of MacroD1 KO mice revealed a reduction in intracellular ADP-ribosylation in all subcellular compartments including mitochondria (Suppl. Fig. 8B). Given that MacroD1 reverses ADP-ribosylation, one would have instead

expected the opposite. However, we strongly believe that this reduction is a consequence of decreased overall NAD⁺ levels (Suppl. Fig. 8D) resulting from the redox shift that takes place following the knockdown of MacroD1 (Fig. 5E). This, together with the fact that a mitochondrial ADP-ribosylhydrolase has not unambiguously been identified makes it technically challenging to identify direct target proteins of MacroD1.

Looking at the mitochondrial proteins that were identified to be ADP-ribosylated and to interact with MacroD1 (Suppl. Fig. 8F, proteins in yellow), one can observe that a lot of enzymes involved in the TCA cycle, its anaplerotic fluxes, and the respiratory chain represent potentially interesting targets. It is intriguing to speculate that in synergy with a potential mitochondrial ADP-ribosyltransferase, MacroD1 senses mitochondrial (and whole cell) NAD⁺ level and adjusts mitochondrial metabolism accordingly by fine-tuning the activity of both TCA cycle and respiratory chain. While we would love to uncover the full cycle, we do believe that this exceeds the scope of this manuscript, as it is technically very challenging and would ideally require the identification of the ADP-ribosyltransferase that opposes MacroD1.

3. The newly added NADPH/NADP⁺ data only show static ratios and do not reveal their dynamic responses under metabolic stress.

Response: We have added this data to support our hypothesis that loss of MacroD1 would result in oxidative stress (Fig. 3E), which the cells try to combat by redirecting glucose into the pentose phosphate pathway to generate redox equivalents and glutathione (Fig. 5A-E). Given that C2C12 and U2OS cells are destined to die upon loss of MacroD1, we do not believe that adding an additional metabolic stress will provide further informative value. Additionally, NADPH/NADP⁺ measurements at later stages after MacroD1 knockdown would be hard to interpret, as cells are already dying. We have added this limitation in the manuscript on page 10.

4. The authors have added experiments on mitochondrial DNA copy numbers, the assessment of mitochondrial quality seems still incomplete. Can further analysis be conducted on mitochondrial membrane potential and the activity of mitochondrial respiratory chain complexes to more comprehensively evaluate changes in mitochondrial function?

Response: We thank the reviewer for the experiment suggestions. However, we would like to draw the reviewer's attention to the fact that we did indeed probe for mitochondrial membrane potential upon loss of MacroD1 (Suppl. Fig. 4D) and could observe a reduction in polarization. Furthermore, we have also already addressed mitochondrial respiration by Seahorse analysis after the knockdown of MacroD1 and report in the manuscript that both basal and maximal respiration is reduced (Fig. 4D). In line with that, our glucose tracing experiments could demonstrate that the flux from glucose into the TCA cycle is reduced (Fig. 4B and C), again suggesting that the cell's overall respiratory capacity is reduced.

5. The authors mentioned the potential role of MacroD1 in cancer but did not delve into its association with the Warburg effect or the therapeutic prospects of combining metabolic inhibitors.

Response: We thank the reviewers for pointing to this interesting aspect. In fact, most publications on MacroD1 focus on its potential role in cancer (reviewed in Feijis *et al.*, *Cancers* **12**(3):604, 2020), which is why we mention this in the manuscript (discussion p. 15) but do not further experimentally address it. In fact, the manuscript aimed to decipher the role of MacroD1 in cell and organismal physiology, hence focusing on muscle where the hydrolase is most expressed. However, as detailed in the discussion, the general observation that MacroD1 expression correlates with proliferation rates and invasiveness of cancer cells is consistent with our hypothesis that MacroD1 promotes oxidative metabolism. Cancer cell migration

often correlates with a more oxidative metabolism, while non-metastatic tumors are frequently characterized by a more glycolytic metabolism (reviewed e.g., in Martínez-Reyes *et al.*, *Nat Rev Cancer* **21**:669 (2021)). While the scope of this manuscript is not to provide experiments that address cancer invasiveness in the presence or absence of MacroD1, we have nevertheless performed lethality experiments to corroborate our hypothesis with actual data.

Fig. 2: Lethality of MacroD1 knockdown in different cell lines. MacroD1 was knocked down in different cell lines and viability was assessed 72h after siRNA transfection by CellTiter-Glo.

We knocked down MacroD1 in the two cell models most frequently used in the study, as well as in MCF7 and MDA-MB-231 cancer cell lines, whereas MCF7 cells are known to be more oxidative and MDA-MB-231 more glycolytic in comparison to each other. As expected, there is a trend towards more pronounced lethality upon knockdown of MacroD1 in those cell models with higher oxidative phosphorylation (Fig. 2 above, not part of the manuscript). Thus, since cancer is not the focus of the paper and this data is in agreement with published literature, we abstain from adding this data to the manuscript.

Minor comments:

1. The resolution of the EM images in Suppl. Fig. 2C is low. It is recommended to provide higher-magnification zoomed-in images to clearly show the abnormal mitochondrial cristae structures.

Response: There seems to be an unclarity for which we would like to apologize. The higher magnification images clearly showing abnormal mitochondrial cristae structure are provided in Fig. 2F.

The images in Suppl. Fig. 2C are representative images that belong to the quantification shown in Fig. 2E and were requested during the first round of revision (reviewer 1, point 5). Suppl. Fig. 2C was left in the manuscript to illustrate the ultrastructural consequences of MacroD1 knockout.

2. Some experiments (e.g., IPGTT/IPITT) do not specify whether multiple comparisons were corrected. The statistical methods need to be clarified.

Response: We thank the reviewer for raising this important point. We have reanalyzed the data sets of Fig. 1D/E and Fig. 6C/D using a two-way Anova, including Geisser-Greenhouse correction, as this test is commonly used for comparing two or more groups over time. Figures, Figure legends, and material and method sections have been corrected and adjusted accordingly (Fig.1D/E, Fig. 6C/D, page 18, 20, 22, and page 30).